# Inner Classifier-Free Guidance and Its Taylor Expansion for Diffusion Models

**Shikun Sun**[1,2]**, Longhui Wei**[3]**, Zhicai Wang**[4]**, Zixuan Wang**[1,2]**, Junliang Xing**[1,2]**,
Jia Jia**[1,2*]**& Qi Tian**[3*]

[1]Tsinghua University, [2]BNRist, [3]Huawei Inc., [4]University of Science and Technology of China
{ssk21,wangzixu21}@mails.tsinghua.edu.cn, weilh2568@gmail.com
wangzhic@mail.ustc.edu.cn, {jlxing, jjia}@tsinghua.edu.cn
tian.qi1@huawei.com

## Abstract

Classifier-free guidance (CFG) is a pivotal technique for balancing the diversity and fidelity of samples in conditional diffusion models. This approach involves utilizing a single model to jointly optimize the conditional score predictor and unconditional score predictor, eliminating the need for additional classifiers. It delivers impressive results and can be employed for continuous and discrete condition representations. However, when the condition is continuous, it prompts the question of whether the trade-off can be further enhanced. Our proposed inner classifier-free guidance (ICFG) provides an alternative perspective on the CFG method when the condition has a specific structure, demonstrating that CFG represents a first-order case of ICFG. Additionally, we offer a second-order implementation, highlighting that even without altering the training policy, our second-order approach can introduce new valuable information and achieve an improved balance between fidelity and diversity for Stable Diffusion.

## 1 Introduction

Diffusion models have garnered significant achievements in tasks involving image and audio generation (Sohl-Dickstein et al., 2015; Ho et al., 2020; Rombach et al., 2022; Podell et al., 2023; Huang et al., 2023; Wang et al., 2024). These models exhibit comparable, and in some cases, superior performance to GAN-based models (Brock et al., 2019) and autoregressive models (Razavi et al., 2019) regarding diversity and fidelity. Notably, text-based image generation models have emerged as particularly successful examples, including Stable Diffusion (Rombach et al., 2022), SDXL (Podell et al., 2023), DALL·E 2 (Ramesh et al., 2022).

There are primarily two approaches to introducing or enhancing guidance in diffusion models: using classifiers (Dhariwal & Nichol, 2021) and employing classifier-free guidance (Ho & Salimans, 2022) (CFG). In the case of classifiers, an external trained classifier is employed to guide the diffusion model at each timestep towards achieving a higher probability according to the classifier's judgment, which is also be extended to energy-based guidance (Zhao et al., 2022; Lu et al., 2023; Sun et al., 2023). On the other hand, classifier-free guidance involves utilizing a single model to optimize the conditional score predictor and unconditional score predictor jointly. The discrepancy between these predictors is then employed as guidance, which is subsequently added to the score function. This approach eliminates the need for additional classifiers and has delivered impressive results (Ho & Salimans, 2022).

Nevertheless, none of these methods impose specific constraints on the condition space, resulting in the underutilization of the benefits of continuity when the condition space is continuous. Our focus lies in exploring whether the nature of continuity can be effectively applied to CFG. Moreover, in the case of widely used text-based image diffusion models like Stable Diffusion (Rombach et al., 2022), the complexity of the text encoder (Radford et al., 2021) raises the question of whether a structured continuous space exists. This space could potentially enhance the balance between fidelity and diversity in generated samples.

---

[*]Corresponding author

To address these concerns, we propose a novel interpretation of CFG. Our approach assumes that the condition space possesses a (local) cone structure. Under this assumption, CFG can be seen as a first-order Taylor expansion of our proposed inner classifier-free guidance (ICFG) method. Building on this interpretation, we further introduce a second-order Taylor expansion of ICFG and reveal an alternative energy-based formulation. Surprisingly, we discover that the second-order Taylor expansion of ICFG yields enhancements for Stable Diffusion, even without modifying the training policy. This finding suggests the existence of a structured continuous condition space that has the potential to enhance the sample performance of Stable Diffusion further.

To summarize, our main contributions are three-fold as follows:

- We introduce ICFG and analyze the convergence of its Taylor expansion under specific conditions.
- We demonstrate that CFG can be regarded as a first-order ICFG and propose a second-order Taylor expansion for our ICFG.
- We apply the second-order ICFG to the Stable Diffusion model and observe that, remarkably, our new formulation yields valuable information and enhances the trade-off between fidelity and diversity, even without modifying the training policy.

## 2 BACKGROUND

### 2.1 DIFFUSION MODELS

Diffusion models encompass various formulations, and we will provide a brief overview of score-based diffusion models (Song & Ermon, 2019; Song et al., 2021b;a; Sun et al., 2023; Ni et al., 2023) (SBDMs) as they offer a solid foundation for guidance methods. SBDMs employ a stochastic differential equation (SDE) to diffuse the data distributions towards known distributions, typically Gaussian distributions. By learning the necessary information to reverse the diffusion process while preserving the marginal distribution, we can sample from the known distribution and subsequently reverse the SDE. This process is equivalent to sampling directly from the data distribution.

Denote the unknown dataset distribution as $q(\mathbf{x}_0)$, where $\mathbf{x}_0 \in \mathbb{R}^d$. We aim to sample from $q(\mathbf{x}_0)$. We also have the terminal distribution $q(\mathbf{x}_T)$, where $\mathbf{x}_T \in \mathbb{R}^d$. To connect these two distributions, we introduce a forward diffusion process $\{\mathbf{x}_t\}_{t \in [0,T]}$, where $q(\mathbf{x}_t)$ or $q_t(\mathbf{x})$ represents the distribution of $\mathbf{x}_t$. We assume that this diffusion process follows a SDE:

$$d\mathbf{x} = \mathbf{f}(\mathbf{x}, t)dt + g(t)d\mathbf{w}, \tag{1}$$

where $\mathbf{f}(\mathbf{x}, t)$ is the drift term, $g(t)$ is the diffusion coefficient, and $\mathbf{w}$ is the standard Wiener process. In the work by Song et al. (2021b), it has been shown that to reverse the SDE while maintaining the marginal distribution, the score function $\mathbf{s}(\mathbf{x}, t)$ is the only required information. The score function is defined as follows:

$$\mathbf{s}(\mathbf{x}, t) = \nabla_{\mathbf{x}_t} \log q(\mathbf{x}_t). \tag{2}$$

Then, the reverse-time SDE is:

$$d\mathbf{x} = [\mathbf{f}(\mathbf{x}, t) - g(t)^2 \mathbf{s}(\mathbf{x}, t)]dt + g(t)d\overline{\mathbf{w}}, \tag{3}$$

where the symbol $\overline{\mathbf{w}}$ represents another standard Wiener process that is independent of the Wiener process $\mathbf{w}$ in the forward diffusion process. The reverse-time SDE is equivalent to the reverse-time diffusion process $\{\mathbf{x}_t\}_{t \in [T,0]}$. Then we can sample from the known distribution $q(\mathbf{x}_T)$, and reverse the SDE to get the sample from the data distribution $q(\mathbf{x}_0)$.

The works of Vincent (2011); Song et al. (2021b) present a feasible method to estimate the score functions of complex datasets using a neural network $\mathbf{s}^\theta(\mathbf{x}, t)$. The optimization objective is defined as follows:

$$\theta^* = \arg\min_\theta \int_0^T \mathbb{E}_{q(\mathbf{x}_0)q_{0t}(\mathbf{x}_t|\mathbf{x}_0)} \left[ \lambda(t) \left\| \mathbf{s}^\theta(\mathbf{x}_t, t) - \nabla_{\mathbf{x}_t} \log q_{0t}(\mathbf{x}_t|\mathbf{x}_0) \right\|^2 \right] dt, \tag{4}$$

where $\lambda(t)$ is a weighting function, and $q_{0t}(\mathbf{x}_t|\mathbf{x}_0)$ is the transition probability from $\mathbf{x}_0$ to $\mathbf{x}_t$.

Furthermore, consider Eq. (1) where we have $\mathbf{x}_t = \alpha_t \mathbf{x}_0 + \beta_t \mathbf{z}$, with $\mathbf{z}$ being a standard Gaussian distribution, and $\alpha_t$ and $\beta_t$ representing the corresponding coefficients. In many scenarios, the diffusion score is parameterized as $\epsilon^\theta(\mathbf{x}, t) = -\beta_t \mathbf{s}^\theta(\mathbf{x}, t)$. Moreover, in the case of conditional diffusion models, where the data $\mathbf{x}$ is accompanied by a conditioning variable $\mathbf{c}$, the only modification is to include $\mathbf{c}$ as an input to $\epsilon^\theta$, resulting in $\epsilon^\theta(\mathbf{x}, \mathbf{c}, t)$.

## 2.2 CLASSIFIER GUIDANCE FOR DIFFUSION MODELS

Dhariwal & Nichol (2021) introduce classifier guidance for diffusion models to enhance control over conditions. Assume that the learned conditional diffusion score is denoted as $\epsilon^\theta(\mathbf{x}, \mathbf{c}, t)$, and we have a set of classifiers $p_t^\theta(\mathbf{c}|\mathbf{x}_t)$ that predict the condition $\mathbf{c}$ based on the diffused data $\mathbf{x}_t$ at various time steps $t$. In this case, we can modify the diffusion score as follows:

$$\widetilde{\epsilon}^\theta(\mathbf{x}_t, \mathbf{c}, t) = \epsilon^\theta(\mathbf{x}_t, \mathbf{c}, t) - w\beta_t \nabla_{\mathbf{x}_t} \log p_t^\theta(\mathbf{c}|\mathbf{x}_t) = -\beta_t \nabla_{\mathbf{x}_t} \left[ \log q^\theta(\mathbf{x}_t|\mathbf{c}) + w \log p_t^\theta(\mathbf{c}|\mathbf{x}_t) \right], \quad (5)$$

where $w$ represents a weighting factor that controls the strength of the guidance. The modified diffusion score, denoted as $\widetilde{\epsilon}^\theta(\mathbf{x}_t, \mathbf{c}, t)$, replaces the original diffusion score $\epsilon^\theta(\mathbf{x}_t, \mathbf{c}, t)$ in the sampling process. At each time step $t$, $\widetilde{\epsilon}^\theta(\mathbf{x}_t, \mathbf{c}, t)$ serves as the diffusion score for a new conditional distribution.

$$\widetilde{q}^\theta(\mathbf{x}_t|\mathbf{c}) \propto q^\theta(\mathbf{x}_t|\mathbf{c}) p_t^\theta(\mathbf{c}|\mathbf{x}_t)^w. \quad (6)$$

It can be observed that $\widetilde{q}^\theta(\mathbf{x}_t|\mathbf{c})$ can be seen as the conditional distribution $q^\theta(\mathbf{x}_t|\mathbf{c})$ multiplied by $p_t^\theta(\mathbf{c}|\mathbf{x}_t)^w$. This indicates that the modified diffusion score assigns a higher probability to the data $\mathbf{x}_t$ that is more likely to be associated with the condition $\mathbf{c}$. Consequently, the modified diffusion score allows for a trade-off between sample diversity and sample fidelity.

The classifier guidance can also be applied to an unconditional diffusion score. For the unconditional diffusion score $\epsilon^\theta(\mathbf{x}, t)$, using the same set of classifiers, the modified diffusion score is given by:

$$\widetilde{\epsilon}^\theta(\mathbf{x}_t, \mathbf{c}, t) = \epsilon^\theta(\mathbf{x}_t, t) - (w+1)\beta_t \nabla_{\mathbf{x}_t} \log p_t^\theta(\mathbf{c}|\mathbf{x}_t) = -\beta_t \nabla_{\mathbf{x}_t} \left[ \log q^\theta(\mathbf{x}_t) + (w+1) \log p_t^\theta(\mathbf{c}|\mathbf{x}_t) \right]. \quad (7)$$

The corresponding guided intermediate distribution is:

$$\widetilde{q}^\theta(\mathbf{x}_t|\mathbf{c}) \propto q^\theta(\mathbf{x}_t) p_t^\theta(\mathbf{c}|\mathbf{x}_t)^{w+1}. \quad (8)$$

While the experiments on guiding the unconditional diffusion score may not have yielded as remarkable results as the conditional case initially (Dhariwal & Nichol, 2021), this formula can still be further connected to the concept of classifier-free guidance.

## 2.3 CLASSIFIER-FREE GUIDANCE FOR DIFFUSION MODELS

To avoid training classifiers, Ho & Salimans (2022) propose an alternative approach called classifier-free guidance (CFG) for diffusion models. The main idea behind CFG is to use a single model to simultaneously fit both the conditional score predictor and the unconditional score predictor. This is achieved by randomly replacing the condition $\mathbf{c}$ with $\varnothing$ (an empty value). By doing so, one can obtain the conditional score predictor $\epsilon^\theta(\mathbf{x}, \mathbf{c}, t)$ and the unconditional score predictor $\epsilon^\theta(\mathbf{x}, t)$, which is equivalent to $\epsilon^\theta(\mathbf{x}, \varnothing, t)$. Then, because

$$\begin{aligned} \nabla_{\mathbf{x}_t} \left[ \log p_t(\mathbf{c}|\mathbf{x}_t) \right] &= \nabla_{\mathbf{x}_t} \left[ \log q(\mathbf{x}_t|\mathbf{c}) - \log q(\mathbf{x}_t) + \log p(\mathbf{c}) \right] \\ &= \nabla_{\mathbf{x}_t} \left[ \log q(\mathbf{x}_t|\mathbf{c}) - \log q(\mathbf{x}_t) \right], \end{aligned} \quad (9)$$

which indicates that after applying the operator $\nabla_{\mathbf{x}_t}$, we can replace the last term of Equation (7) with $\log q^\theta(\mathbf{x}_t|\mathbf{c}) - \log q^\theta(\mathbf{x}_t)$ to achieve a similar effect. Then we get the enhanced diffusion score:

$$\begin{aligned} \hat{\epsilon}^\theta(\mathbf{x}_t, \mathbf{c}, t) &= (w+1)\epsilon^\theta(\mathbf{x}_t, \mathbf{c}, t) - w\epsilon^\theta(\mathbf{x}_t, t) \\ &= -\beta_t \nabla_{\mathbf{x}_t} \left[ \log q^\theta(\mathbf{x}_t|\mathbf{c}) + w(\log q^\theta(\mathbf{x}_t|\mathbf{c}) - \log q^\theta(\mathbf{x}_t)) \right] \quad (10) \\ &= -\beta_t \nabla_{\mathbf{x}_t} \left[ \log q^\theta(\mathbf{x}_t) + (w+1)(\log q^\theta(\mathbf{x}_t|\mathbf{c}) - \log q^\theta(\mathbf{x}_t)) \right], \end{aligned}$$

whose enhanced intermediate distribution is:

$$\hat{q}^\theta(\mathbf{x}_t|\mathbf{c}) \propto q^\theta(\mathbf{x}_t) \left[ \frac{q^\theta(\mathbf{x}_t|\mathbf{c})}{q^\theta(\mathbf{x}_t)} \right]^{w+1}. \quad (11)$$

It is worth noting that both of these guidance methods can be regarded as a more general form of energy-based guidance (Zhao et al., 2022; Lu et al., 2023). In this case, the formulation of the intermediate time distribution is:

$$\overline{q}^\theta(\mathbf{x}_t|\mathbf{c}) \propto q^\theta(\mathbf{x}_t)e^{-(w+1)\beta(\mathbf{x}_t)}, \qquad (12)$$

where $\beta(\mathbf{x}_t)$ is an arbitrary energy function.

## 3 INNER CLASSIFIER-FREE GUIDANCE

Firstly, consider the enhanced intermediate distribution $\overline{q}^\theta(\mathbf{x}_t|\mathbf{c})$ obtained by Eq. (8) or Eq. (11), given the condition $\mathbf{c}$. The question is whether these enhanced distributions follow the same diffusion forward process as the original intermediate distribution $q^\theta(\mathbf{x}_t)$. We have:

**Theorem 3.1.** *Given condition $\mathbf{c}$, the enhanced transition kernel $\overline{q}_{0t}^\theta(\mathbf{x}_t|\mathbf{x}_0, \mathbf{c})$ by Eq. (8) or Eq. (11) equals to the original transition kernel $q_{0t}^\theta(\mathbf{x}_t|\mathbf{x}_0, \mathbf{c}) = q_{0t}^\theta(\mathbf{x}_t|\mathbf{x}_0)$ does not hold trivially. Specifically, when $w = 0$, the equation holds.*

The proof and discussions of Theorem 3.1 is in the Appendix A. Theorem 3.1 suggests that in the majority of cases, the enhanced intermediate distribution $\overline{q}^\theta(\mathbf{x}_t|\mathbf{c})$ at different timesteps $t$ does not adhere to the same SDE as the original intermediate distribution $q^\theta(\mathbf{x}_t)$ or the original conditional intermediate distribution $q^\theta(\mathbf{x}_t|\mathbf{c})$, which is also mentioned in the work of Lu et al. (2023); Du et al. (2023).

There are two important clarifications to make regarding Theorem 3.1:

- $q^\theta(\mathbf{x}_0|\mathbf{x}_t)$ appears to never be a $\delta$ distribution. However, under certain conditions, such as a low noise level or when $\mathbf{x}_0$ is sparse, it can be approximately close to a $\delta$ distribution.

- Even if $\overline{q}_{0t}^\theta(\mathbf{x}_t|\mathbf{x}_0, \mathbf{c})$ deviates from the original diffusion process, The sampling process may still be effective based on the underlying principles of Langevin dynamics.

To address this issue, we incorporate the guidance strength $w$ and represent the enhanced intermediate distribution in a more clear form as $q^\theta(\mathbf{x}_t|\mathbf{c}, w + 1)$. For instance, in the context of CFG, we can express $q^\theta(\mathbf{x}_t|\mathbf{c}, w + 1) \propto q^\theta(\mathbf{x}_t)\left[\frac{q^\theta(\mathbf{x}_t|\mathbf{c})}{q^\theta(\mathbf{x}_t)}\right]^{w+1}$.

Let $\beta = w + 1$. It is worth noting that in Theorem 3.1, when $\beta = 1$, we have $q_{0t}^\theta(\mathbf{x}_t|\mathbf{x}_0, \mathbf{c}, \beta) = q_{0t}^\theta(\mathbf{x}_t|\mathbf{x}_0)$. The question arises: Can we always ensure that $\beta = 1$? In other words, can we treat $\beta$ and $\mathbf{c}$ as the same variable, such that $q_{0t}^\theta(\mathbf{x}_t|\mathbf{x}_0, \mathbf{c}, \beta) = q_{0t}^\theta(\mathbf{x}_t|\mathbf{x}_0)$ holds consistently? By doing so, we incorporate the condition strength into the condition variable itself, and we refer to this approach as inner classifier-free guidance (ICFG).

To establish a well-defined ICFG, it is necessary to impose certain structural assumptions on the space $\mathcal{C}$ of the condition $\mathbf{c}$. The following assumptions are made:

**Assumption 3.1.**

- $\mathcal{C}$ *is a cone, which means* $\forall \beta \in \mathbb{R}^+, \forall \mathbf{c} \in \mathcal{C}, \beta\mathbf{c} \in \mathcal{C}$.

- *For each* $\mathbf{c} \in \mathcal{C}$, $\|\mathbf{c}\|$ *represents the guidance strength and* $\frac{\mathbf{c}}{\|\mathbf{c}\|}$ *represents the guidance direction.*

For implementation purposes, as the origin $\mathcal{C}$ does not inherently form a cone, it is common practice to extend the existing meaningful space $\mathcal{C}$ to conform to a cone structure. This extended space is denoted as $\overline{\mathcal{C}} = \{\mathbf{c}_0 + \beta(\mathbf{c} - \mathbf{c}_0)|\beta \in \mathbb{R}^+, \mathbf{c} \in \mathcal{C}\}$, where the vertex of the cone $\mathbf{c}_0$ is not necessarily $\mathbf{0}$. Further details can be found in Appendix D.

Under Assumption 3.1, we define $\overline{q}^\theta(x_t|c) = q^\theta(\mathbf{x}_t|\mathbf{c}, \beta) \triangleq q^\theta(\mathbf{x}_t|\beta\mathbf{c})$. Based on this definition, we can state the following Corollary 3.1.1:

**Corollary 3.1.1.** *Given condition $\mathbf{c}$ and the guidance strength $\beta = w + 1$, we have:*

$$q_{0t}^\theta(\mathbf{x}_t|\mathbf{x}_0, \mathbf{c}, \beta) = q_{0t}^\theta(\mathbf{x}_t|\mathbf{x}_0).$$

Corollary 3.1.1 indicates that for ICFG, the forward diffusion process consistently remains the same as the original forward diffusion process, given condition $\mathbf{c}$ and guidance strength $\beta$.

We propose a training policy that allows the model to assess the guidance strength based on a correlation metric $r(\mathbf{x}, \mathbf{c})$, which measures the relationship between the condition and the data. The training policy is outlined in Algorithm 1.

---

**Algorithm 1** Training policy for ICFG

---

**Require:** $r(\mathbf{x}, \mathbf{c})$: similarity metric
**Require:** $p_{uncond}$: probability of unconditional training
 1: $\overline{r} = \mathbb{E}_{\mathbf{x},\mathbf{c}} r(\mathbf{x}, \mathbf{c})$
 2: **repeat**
 3:     $(\mathbf{x}, \mathbf{c}) \sim p(\mathbf{x}, \mathbf{c})$
 4:     $\overline{\mathbf{c}} = \mathbf{c}/r(\mathbf{x}, \mathbf{c}) * \overline{r}$
 5:     $\overline{\mathbf{c}} \leftarrow \varnothing$ with probability $p_{uncond}$
 6:     $t \sim U[0, T]$
 7:     $\epsilon \sim \mathcal{N}(\mathbf{0}, \mathbf{I})$
 8:     $\mathbf{x}_t = \alpha_t \mathbf{x}_0 + \beta_t \epsilon$
 9:     Take gradient step on $\nabla_\theta \|\epsilon^\theta(\mathbf{x}_t, \overline{\mathbf{c}}) - \epsilon\|^2$
10: **until** converged

---

## 4 TAYLOR EXPANSION OF ICFG AND ITS CONVERGENCE

In most cases, we use the extended cone $\overline{\mathcal{C}} = \{\beta\mathbf{c} | \beta \in \mathbb{R}^+, \mathbf{c} \in \mathcal{C}\}$. Consequently, we need to consider the Taylor expansion of ICFG at $\beta = 1$ to estimate the score under conditions within $\overline{\mathcal{C}}/\mathcal{C}$. The n-th order Taylor expansion of $\overline{\epsilon}^\theta(\mathbf{x}_t | \beta\mathbf{c})$ at $\beta = 1$ is given by:

$$\overline{\epsilon}^\theta(\mathbf{x}_t | \beta\mathbf{c}) = \overline{\epsilon}^\theta(\mathbf{x}_t | \mathbf{c}) + \sum_{k=1}^n \frac{1}{k!} \left. \frac{\partial^k \overline{\epsilon}^\theta(\mathbf{x}_t | \beta\mathbf{c})}{\partial \beta^k} \right|_{\beta=1} (\beta - 1)^k + R_n(\beta), \tag{13}$$

where $R_n(\beta)$ represents the remainder term. It is evident that CFG is a first-order Taylor expansion of ICFG at $\beta = 1$ without $R_1(\beta)$ and with the following estimation:

$$\left. \frac{\partial \overline{\epsilon}^\theta(\mathbf{x}_t | \beta\mathbf{c})}{\partial \beta} \right|_{\beta=1} \approx \overline{\epsilon}^\theta(\mathbf{x}_t | \mathbf{c}) - \overline{\epsilon}^\theta(\mathbf{x}_t | \mathbf{0}) = \overline{\epsilon}^\theta(\mathbf{x}_t | \mathbf{c}) - \overline{\epsilon}^\theta(\mathbf{x}_t) \tag{14}$$

Then we will discuss the convergence of the Taylor expansion of ICFG in Eq. (13).

Firstly, considering this problem from the model space $\mathcal{S} = \{\overline{\epsilon}^\theta(\mathbf{x}_t | \beta\mathbf{c}) | \forall\theta, \forall\mathbf{x}, \forall\mathbf{c}\}$, if we judiciously choose the components of the neural network such that the function $\overline{\epsilon}^\theta(\mathbf{x}_t | \beta\mathbf{c})$ becomes analytic with respect to $\beta$, then the convergence of the Taylor expansion near $\beta = 1$ is guaranteed trivially.

Secondly, if we desire a bound for $R_n(\beta)$ for any $\beta \in [0, B]$, certain assumptions must be introduced. Before that, consider the specific quantitative relationships of Stable Diffusion, as depicted in Figure 1. It becomes apparent that the estimation of $\left. \frac{\partial \overline{\epsilon}^\theta(\mathbf{x}_t | \beta\mathbf{c})}{\partial \beta} \right|_{\beta=1}$ is relatively small. Consequently, we propose the following Assumption:

**Assumption 4.1.** *For all $k \in \mathbb{N}^+$, the k-th order partial derivative $\frac{\partial^k \overline{\epsilon}^\theta(\mathbf{x}_t | \beta\mathbf{c})}{\partial \beta^k}$ at $\beta \in [0, B]$ is bounded by $M_k$.*

Then, we have:

**Theorem 4.1.** *Under Assumption 4.1, the remainder term $R_n(\beta)$ of the n-th order taylor expansion of $\overline{\epsilon}^\theta(\mathbf{x}_t | \beta\mathbf{c})$ at $\beta = 1$ is bounded by $\frac{M_{n+1}}{(n+1)!} B^{n+1}$. This bound converges to 0 when the sequence $M_{n+1} \sim o\left(\sqrt{n+1} \left[\frac{n+1}{eB}\right]^{n+1}\right)$.*

By Stirling's formula, the proof will be discussed in the Appendix B.

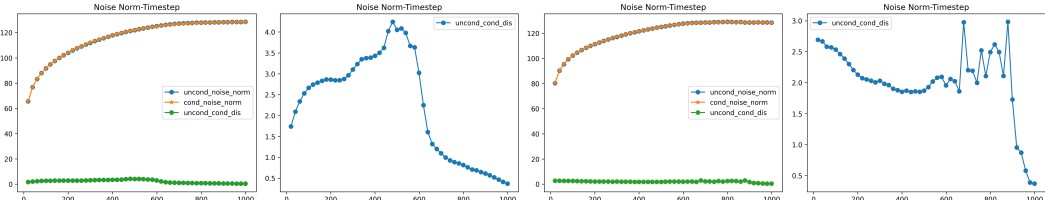

Figure 1: The quantitative relationships of Stable Diffusion during sampling. These figures show the relationship between the norm of the predicted score $\bar{\epsilon}^\theta(\mathbf{x}_t|\mathbf{c})$, $\bar{\epsilon}^\theta(\mathbf{x}_t)$ and the distance norm between them during the 50 sampling timesteps of CFG. The guidance strength for the first two figures is set to 3, while the last two figures have a guidance strength of 80. The caption for all figures states, "A photograph of an astronaut riding a horse."

## 5   IMPLEMENTATION OF SECOND-ORDER ICFG FOR STABLE DIFFUSION

For the pretrained popular Stable Diffusion, we present two approaches to implement the second-order ICFG. The first method follows a straightforward Taylor expansion, as outlined in Algorithm 2. Notably, when we use $\frac{y_2-y_1}{x_2-x_1}$ of two points $(x_1, y_1), (x_2, y_2)$ to estimate the gradient at $\frac{x_1+x_2}{2}$, the second-order term is unique. Further details will be discussed in the Appendix C.

---

**Algorithm 2** Strict sample algorithm for second-order ICFG

---

**Require:** $m$: middle point for estimate second-order term
**Require:** $w$: guidance strength on conditional score predictor
**Require:** $\mathbf{c}$: condition for sampling
**Require:** Require $\{t_1, t_2, ..., t_N\}$ increasing timestep sequence of sampling
**Require:** $Sample(\mathbf{z}_t, \epsilon_t)$: sample algorithm for diffusion models given $\mathbf{z}_t$ and $\epsilon_t$
1: $\mathbf{z}_N \sim \mathcal{N}(\mathbf{0}, \mathbf{I})$
2: **for** $i = N, ..., 1$ **do**
3:     $\bar{\epsilon}_t = \epsilon^\theta(\mathbf{z}_i, \mathbf{c}) + w(\epsilon^\theta(\mathbf{z}_i, \mathbf{c}) - \epsilon^\theta(\mathbf{z}_i))$
        $+ w^2 \frac{1}{m(1-m)}\left((1-m)\epsilon^\theta(\mathbf{z}_i) + m\epsilon^\theta(\mathbf{z}_i, \mathbf{c}) - \epsilon^\theta(\mathbf{z}_i, m\mathbf{c})\right)$
4:     $\mathbf{z}_{i-1} = Sample(\mathbf{z}_i, \bar{\epsilon}_t)$
5: **end for**
6: **return** $\mathbf{z}_0$

---

However, in practical scenarios, the second-order term may suffer from a large bias due to the unchanged training policy, and this bias is further amplified by the coefficient $w^2$. To address this issue, we propose an alternative approach called the non-strict sample algorithm, presented in Algorithm 3. This algorithm offers a practical solution and can be effectively applied to mitigate the aforementioned problem.

The only distinction between Algorithm 3 and Algorithm 2 lies in assigning a completely unrestricted hyperparameter $v$ to the second-order term. This modification allows for more flexible control over the second-order term.

## 6   EXPERIMENTS

The primary motivation behind our experiments is twofold. Firstly, we seek to showcase the efficacy of our new sampling algorithm in leveraging the continuity of $\mathcal{C}$ and incorporating valuable information to achieve an improved balance between diversity and fidelity, even in cases where $\mathcal{C}$ does not exhibit a "cone" structure apparently. Secondly, we aim to demonstrate that our new training policy enables the model to capture better the inherent "cone" structure of $\mathcal{C}$. To accomplish the first objective, we utilize the pretrained Stable Diffusion v1.5 model and apply our Algorithm 3 to generate sampled images. More implementation details are in Appendix D. Nevertheless, our second-order ICFG continues to demonstrate its advantages in these experiments. For the second target, we ex-

---

**Algorithm 3** Non-strict sample algorithm for second-order ICFG

---

**Require:** $m$: middle point for estimate second-order term
**Require:** $w$: first-order guidance strength on conditional score predictor
**Require:** $v$: second-order guidance strength on conditional score predictor
**Require:** $\mathbf{c}$: condition for sampling
**Require:** Require $\{t_1, t_2, ..., t_N\}$ increasing timestep sequence of sampling
**Require:** $Sample(\mathbf{z}_t, \epsilon_t)$: sample algorithm for diffusion models given $\mathbf{z}_t$ and $\epsilon_t$
  1: $\mathbf{z}_N \sim \mathcal{N}(\mathbf{0}, \mathbf{I})$
  2: **for** $i = N, ..., 1$ **do**
  3:    $\bar{\epsilon}_t = \epsilon^\theta(\mathbf{z}_i, \mathbf{c}) + w(\epsilon^\theta(\mathbf{z}_i, \mathbf{c}) - \epsilon^\theta(\mathbf{z}_i))$
         $+v\frac{1}{m(1-m)}\left((1-m)\epsilon^\theta(\mathbf{z}_i) + m\epsilon^\theta(\mathbf{z}_i, \mathbf{c}) - \epsilon^\theta(\mathbf{z}_i, m\mathbf{c})\right)$
  4:    $\mathbf{z}_{i-1} = Sample(\mathbf{z}_i, \bar{\epsilon}_t)$
  5: **end for**
  6: **return** $\mathbf{z}_0$

---

Table 1: The results of varying the guidance strength and the condition space on the MS-COCO validation set.

| Model&Settings | FID ↓ | CLIP Score (%)↑ |
|:---:|:---:|:---:|
| CFG | | |
| $w = 1.0$ | 17.24 | 25.03 |
| $w = 2.0$ | **15.42** | 25.80 |
| $w = 3.0$ | 16.68 | 26.12 |
| $w = 4.0$ | 18.18 | 26.34 |
| $w = 5.0$ | 19.53 | **26.45** |
| Ours | $v = 0.25/0.5/1.0$ | |
| $w = 1.0, \mathcal{C} = \mathcal{C}_{\text{all}}$ | 16.40/17.34/20.70 | 25.46/25.71/25.86 |
| $w = 2.0, \mathcal{C} = \mathcal{C}_{\text{all}}$ | 15.28/15.42/16.34 | 26.11/26.30/26.52 |
| $w = 3.0, \mathcal{C} = \mathcal{C}_{\text{all}}$ | 16.59/16.69/16.88 | 26.37/26.54/26.73 |
| $w = 4.0, \mathcal{C} = \mathcal{C}_{\text{all}}$ | 17.98/18.06/18.20 | 26.51/26.64/26.81 |
| $w = 5.0, \mathcal{C} = \mathcal{C}_{\text{all}}$ | 19.35/19.32/19.45 | 26.59/26.69/**26.86** |
| $w = 1.0, \mathcal{C} = \mathcal{C}_{\text{nouns}}$ | 16.33/17.24/21.78 | 25.02/24.88/24.23 |
| $w = 2.0, \mathcal{C} = \mathcal{C}_{\text{nouns}}$ | **15.22**/15.23/15.71 | 25.86/25.81/25.61 |
| $w = 3.0, \mathcal{C} = \mathcal{C}_{\text{nouns}}$ | 16.59/16.60/16.61 | 26.19/26.18/26.09 |
| $w = 4.0, \mathcal{C} = \mathcal{C}_{\text{nouns}}$ | 18.08/18.07/18.02 | 26.36/26.37/26.30 |
| $w = 5.0, \mathcal{C} = \mathcal{C}_{\text{nouns}}$ | 19.33/19.31/19.47 | 26.48/26.49/26.44 |

clusively fine-tune the U-Net (Ronneberger et al., 2015) of Stable Diffusion v1.5 (Rombach et al., 2022) with Low-Rank Adaptation (Hu et al., 2022; Ruiz et al., 2023).

## 6.1 SAMPLING WITH SECOND-ORDER ICFG

We employ the pretrained Stable Diffusion v1.5 model directly and utilize our Algorithm 3 to generate sampled images. The settings we follow are consistent with those provided in the official repository of Stable Diffusion v1.5. The sampling algorithm employed is PNDM (Liu et al., 2022), and the default number of timesteps is 50. The evaluation of the results, presented in Table 1, Table 2 and Table 3 is based on two metrics: the Fréchet Inception Distance (FID) (Heusel et al., 2017) and the CLIP Score (Radford et al., 2021). The FID metric is calculated by comparing 10,000 generated images with the MS-COCO (Lin et al., 2014) validation dataset, measuring the distance between the distribution of generated images and the distribution of the validation dataset. On the other hand, the CLIP Score is computed between the 10,000 generated images and their corresponding captions by the model ViT-L/14 (Radford et al., 2021), reflecting the similarity between the images and the textual descriptions. In our tables, the default configuration is set to $w = 2.0, v = 0.25, m = 1.1$. We systematically vary the guidance strength $w$ and $v$, the middle point $m$, and the condition space $\mathcal{C}$ to observe the impact of the second-order term. The designs of $\mathcal{C}_{\text{all}}$ and $\mathcal{C}_{\text{nouns}}$ are in Appendix D.

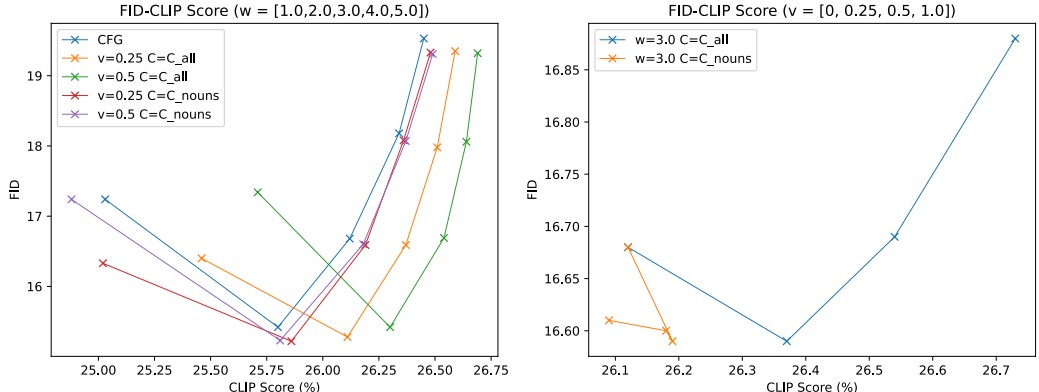

Figure 2: The FID-CLIP Score of varying $w$, $v$ and $\mathcal{C}$.

Table 2: The results of varying the middle points on the MS-COCO validation set. Here $v = 0.5, \mathcal{C} = \mathcal{C}_{\text{all}}$.

| Middle points | FID $\downarrow$ | CLIP Score (%)$\uparrow$ |
|:---:|:---:|:---:|
| $m = 0.5$ | 15.59 | 26.02 |
| $m = 0.8$ | 15.58 | 26.28 |
| $m = 0.9$ | 15.54 | 26.27 |
| $m = 1.05$ | 15.47 | **26.31** |
| $m = 1.1$ | **15.42** | 26.30 |
| $m = 1.2$ | 15.43 | 26.28 |

### 6.1.1 Varying the guidance strength and the space of condition

Here we experimentally validate the first primary claim in this paper: that second-order ICFG can improve the balance between FID and CLIP Score. By varying the guidance strength and space of condition, We determine that the optimal balance between FID and CLIP Score is achieved at $w = 2.0, v = 0.25$. The best $w$ is the same as many other diffusion models (Bao et al., 2023b) Beyond this point, we observe a discernible trade-off between FID and CLIP Score as $w$ increases. In the condition space $\mathcal{C}_{\text{all}}$, we note the presence of a trade-off between FID and CLIP Score as $v$ increases. However, such a trade-off is not evident in the conditional space $\mathcal{C}_{\text{nouns}}$. Additionally, we observe that while the best FID score is obtained in $\mathcal{C}_{\text{nouns}}$, a superior balance between FID and CLIP Score is achieved in $\mathcal{C}_{\text{all}}$. This finding suggests that the condition space $\mathcal{C}_{\text{all}}$ exhibits a more favorable "cone" structure for processing.

### 6.1.2 Varying the middle points

One of the key hyperparameters in the second-order ICFG is the selection of middle point, which is utilized to estimate the second-order term. Two primary factors influence the outcome in this regard. Firstly, if the chosen points are too close to each other, the estimated second-order term fails to capture long-term changes adequately. Secondly, if the middle points are relatively distant from either 0 or 1, the model struggles to estimate the corresponding score. The observed "U" shape of the FID results presented in Table 2 serves to validate our analysis.

### 6.1.3 Varying the number of sampling steps

The number of sampling steps also influences the quality of the generated samples. We have observed that the CLIP Score remains relatively stable across different numbers of sampling steps, while the FID score improves as the number of sampling steps increases. This suggests that the sample quality improves with an increase in the number of sampling steps. However, it is worth

Table 3: The results of varying the sampling steps on the MS-COCO validation set.

| Model&Settings | FID ↓ | CLIP Score (%)↑ |
|:---:|:---:|:---:|
| Ours | $\mathcal{C} = \mathcal{C}_{\text{all}}/\mathcal{C}_{\text{nouns}}$ | |
| $T = 10$ | 15.80/15.86 | 26.13/25.87 |
| $T = 20$ | 15.39/15.40 | **26.15**/25.86 |
| $T = 30$ | 15.29/15.28 | 26.13/25.86 |
| $T = 40$ | 15.29/15.23 | 26.11/25.86 |
| $T = 50$ | 15.28/**15.22** | 26.11/25.86 |

noting that even with a small number of sampling steps, the initial matching degree between the generated text and the image is already quite good.

## 6.2 FEW-SHOT FINE-TUNING FOR STABLE DIFFUSION

To validate the efficacy of our training algorithm, we employ a fine-tuning process on the pretrained Stable Diffusion v1.5 model using Algorithm 1. We then compare the outcomes with those obtained through traditional fine-tuning. By generating cases with varying inner $\beta$ values, we aim to assess the capacity of the model to capture the inherent "cone" structure of $\mathcal{C}$. The results, presented in the Appendix E, demonstrate that our training algorithm yields improved tolerance when coupled with more substantial inner guidance.

## 7 DISCUSSION

ICFG offers a novel perspective for comprehending CFG and can be seen as an extension of CFG. One significant advantage of our ICFG approach is its simplicity in implementation. Furthermore, integrating second-order ICFG into complex conditions in trained diffusion models is straightforward, involving adding a few lines of code. By selecting a suitable space to exploit continuity, we can effectively implement second-order ICFG. In cases where the condition space $\mathcal{C}$ exhibits a well-defined structure, extending the second-order ICFG to higher-order ICFG becomes feasible.

We also offer an intuitive explanation of how the Taylor expansion operates. When we extend the CFG to a higher-order Taylor expansion of ICFG, the corresponding enhanced transition kernel $\bar{q}_{0t}^{\theta}(\mathbf{x}_t|\mathbf{x}_0, \mathbf{c})$ gradually aligns with the original transition kernel more smoothly. In simpler terms, the enhanced transition kernel $\bar{q}_{0t}^{\theta}(\mathbf{x}_t|\mathbf{x}_0, \mathbf{c})$ becomes increasingly similar to the original transition kernel $q_{0t}^{\theta}(\mathbf{x}_t|\mathbf{x}_0)$. This explains why the second-order ICFG can enhance the FID and CLIP Score balance.

Despite its advantages, ICFG also has a few potential disadvantages. Firstly, the training policy of ICFG relies on more precise data pairs to accurately capture the guidance strength during training. However, this requirement can be alleviated by incorporating the similarity function $r(\mathbf{x}, \mathbf{c})$. Secondly, the second-order ICFG necessitates three forward passes of the diffusion model to estimate the second-order term, which can lead to increased sampling time. Nevertheless, this issue can be mitigated by reusing the points for the pre-order term, thereby reducing the computational overhead.

## 8 CONCLUSION

We introduce ICFG, a novel perspective on CFG. ICFG is an extension of CFG and can be readily implemented in training and sampling processes. We further propose the Taylor expansion of ICFG and analyze its convergence properties. By incorporating second-order ICFG, we mitigate the mismatch issue in the diffusion process that arises from CFG. Through our experiments on Stable Diffusion, we validate the efficacy of our second-order approach. In future work, we intend to investigate higher-order ICFG and anticipate further investigations into the application of ICFG in a wide array of diffusion models across diverse data modalities.

## 9 ACKNOWLEDGEMENTS

This work is supported by the National Key R&D Program of China under Grant No.2021QY1500.

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

# A PROOF OF THEOREM 3.1 AND COROLLARY 3.1.1

## A.1 PROOF OF THEOREM 3.1

*Proof.* Firstly, we ignore the symbol $\theta$. Classifier guidance and CFG with guidance strength $w$ have the following enhanced conditional probability:

$$\overline{q}(\mathbf{x}_t|\mathbf{c}) = \frac{1}{Z_t}q(\mathbf{x}_t)q(\mathbf{c}|\mathbf{x}_t)^{w+1}, \tag{15}$$

where $Z_t = \int q(\mathbf{x}_t)q(\mathbf{c}|\mathbf{x}_t)^{w+1}\mathrm{d}\mathbf{x}_t$.

Suppose the enhanced transition kernel $\overline{q}_{0t}(\mathbf{x}_t|\mathbf{x}_0, \mathbf{c})$ equals the original transition kernel $q_{0t}(\mathbf{x}_t|\mathbf{x}_0)$, we have:

$$
\begin{aligned}
\overline{q}_t(\mathbf{x}_t|\mathbf{c}) &= \int \overline{q}_{0t}(\mathbf{x}_t|\mathbf{x}_0, \mathbf{c})\overline{q}_0(\mathbf{x}_0|\mathbf{c})\mathrm{d}\mathbf{x}_0 \\
&= \int q_{0t}(\mathbf{x}_t|\mathbf{x}_0)\overline{q}_0(\mathbf{x}_0|\mathbf{c})\mathrm{d}\mathbf{x}_0 \\
&= \frac{1}{Z_0}\int q_{0t}(\mathbf{x}_t|\mathbf{x}_0)q_0(\mathbf{x}_0)q(\mathbf{c}|\mathbf{x}_0)^{w+1}\mathrm{d}\mathbf{x}_0 \\
&= \frac{1}{Z_0}\int q(\mathbf{x}_t, \mathbf{x}_0)q(\mathbf{c}|\mathbf{x}_0)^{w+1}\mathrm{d}\mathbf{x}_0 \\
&= \frac{1}{Z_0}q(\mathbf{x}_t)\int q(\mathbf{x}_0|\mathbf{x}_t)q(\mathbf{c}|\mathbf{x}_0)^{w+1}\mathrm{d}\mathbf{x}_0 \\
&= \frac{1}{Z_t}q(\mathbf{x}_t)q(\mathbf{c}|\mathbf{x}_t)^{w+1}.
\end{aligned}
\tag{16}
$$

Because $q(\mathbf{c}|\mathbf{x}_t) = \int q(\mathbf{x}_0|\mathbf{x}_t)q(\mathbf{c}|\mathbf{x}_0)\mathrm{d}\mathbf{x}_0$. We take the last two terms and then get the following equation:

$$
\begin{aligned}
&\frac{1}{Z_0}\int q(\mathbf{x}_0|\mathbf{x}_t)q(\mathbf{c}|\mathbf{x}_0)^{w+1}\mathrm{d}\mathbf{x}_0 = \frac{1}{Z_t}\left[\int q(\mathbf{x}_0|\mathbf{x}_t)q(\mathbf{c}|\mathbf{x}_0)\mathrm{d}\mathbf{x}_0\right]^{w+1}. \\
\Leftrightarrow\quad &\frac{\int q(\mathbf{x}_0|\mathbf{x}_t)q(\mathbf{c}|\mathbf{x}_0)^{w+1}\mathrm{d}\mathbf{x}_0}{\left[\int q(\mathbf{x}_0|\mathbf{x}_t)q(\mathbf{c}|\mathbf{x}_0)\mathrm{d}\mathbf{x}_0\right]^{w+1}} = \frac{Z_0}{Z_t} \\
\Leftrightarrow\quad &\frac{\mathbb{E}_{\mathbf{x}_0\sim q(\mathbf{x}_0|\mathbf{x}_t)}q(\mathbf{c}|\mathbf{x}_0)^{w+1}}{\left[\mathbb{E}_{\mathbf{x}_0\sim q(\mathbf{x}_0|\mathbf{x}_t)}q(\mathbf{c}|\mathbf{x}_0)\right]^{w+1}} = \frac{Z_0}{Z_t}.
\end{aligned}
\tag{17}
$$

To enhance clarity, let's consider a straightforward scenario. Suppose $\mathbf{x}_0$ comprises only $\mathbf{x}_0^1$ with label $\mathbf{c}^1$ and $\mathbf{x}_0^2$ with label $\mathbf{c}^2$. Given $\mathbf{c} = \mathbf{c}^1$, the left side of Eq. (17) is as follows:

$$
\begin{aligned}
L &= \frac{\frac{1}{e^{\|-\mathbf{x}_t-\alpha_t\mathbf{x}_0^1\|^2/(\beta_t^2)}+e^{\|-\mathbf{x}_t-\alpha_t\mathbf{x}_0^2\|^2/(\beta_t^2)}}\left(e^{-\|\mathbf{x}_t-\alpha_t\mathbf{x}_0^1\|^2/(\beta_t^2)}q(\mathbf{c}^1|\mathbf{x}_0^1)^{w+1}+e^{-\|\mathbf{x}_t-\alpha_t\mathbf{x}_0^2\|^2/(\beta_t^2)}q(\mathbf{c}^1|\mathbf{x}_0^2)^{w+1}\right)}{\left[\frac{1}{e^{\|-\mathbf{x}_t-\alpha_t\mathbf{x}_0^1\|^2/(\beta_t^2)}+e^{\|-\mathbf{x}_t-\alpha_t\mathbf{x}_0^2\|^2/(\beta_t^2)}}\left(e^{-\|\mathbf{x}_t-\alpha_t\mathbf{x}_0^1\|^2/(\beta_t^2)}q(\mathbf{c}^1|\mathbf{x}_0^1)+e^{-\|\mathbf{x}_t-\alpha_t\mathbf{x}_0^2\|^2/(\beta_t^2)}q(\mathbf{c}^1|\mathbf{x}_0^2))\right]^{w+1}} \\
&= \frac{\frac{1}{e^{\|-\mathbf{x}_t-\alpha_t\mathbf{x}_0^1\|^2/(\beta_t^2)}+e^{\|-\mathbf{x}_t-\alpha_t\mathbf{x}_0^2\|^2/(\beta_t^2)}}\left(e^{-\|\mathbf{x}_t-\alpha_t\mathbf{x}_0^1\|^2/(\beta_t^2)}\times 1+e^{-\|\mathbf{x}_t-\alpha_t\mathbf{x}_0^2\|^2/(\beta_t^2)}\times 0\right)}{\left[\frac{1}{e^{\|-\mathbf{x}_t-\alpha_t\mathbf{x}_0^1\|^2/(\beta_t^2)}+e^{\|-\mathbf{x}_t-\alpha_t\mathbf{x}_0^2\|^2/(\beta_t^2)}}\left(e^{-\|\mathbf{x}_t-\alpha_t\mathbf{x}_0^1\|^2/(\beta_t^2)}\times 1+e^{-\|\mathbf{x}_t-\alpha_t\mathbf{x}_0^2\|^2/(\beta_t^2)}\times 0)\right]^{w+1}} \\
&= \left[\frac{e^{\|-\mathbf{x}_t-\alpha_t\mathbf{x}_0^1\|^2/(\beta_t^2)}+e^{\|-\mathbf{x}_t-\alpha_t\mathbf{x}_0^2\|^2/(\beta_t^2)}}{e^{\|-\mathbf{x}_t-\alpha_t\mathbf{x}_0^1\|^2/(\beta_t^2)}}\right]^w,
\end{aligned}
\tag{18}
$$

which is a function of $\mathbf{x}_t$.

For a more general situation: We have N data pairs $(\mathbf{x}_0^i, \mathbf{c}^i)$. Given $\mathbf{c} = \mathbf{c}^1$, then the left side of Eq. (17) is

$$
\begin{aligned}
L &= \frac{\frac{1}{\sum_{i=1}^N e^{\|-\mathbf{x}_t - \alpha_t \mathbf{x}_0^i\|^2/(\beta_t^2)}} \left( \sum_{i=1}^N e^{-\|\mathbf{x}_t - \alpha_t \mathbf{x}_0^i\|^2/(\beta_t^2)} q(\mathbf{c}^1|\mathbf{x}_0^i)^{w+1} \right)}{\left[ \frac{1}{\sum_{i=1}^N e^{\|-\mathbf{x}_t - \alpha_t \mathbf{x}_0^i\|^2/(\beta_t^2)}} \left( \sum_{i=1}^N e^{-\|\mathbf{x}_t - \alpha_t \mathbf{x}_0^i\|^2/(\beta_t^2)} q(\mathbf{c}^1|\mathbf{x}_0^i)^{w+1} \right) \right]^{w+1}} \\
&= \frac{\frac{1}{\sum_{i=1}^N e^{\|-\mathbf{x}_t - \alpha_t \mathbf{x}_0^i\|^2/(\beta_t^2)}} \left( e^{-\|\mathbf{x}_t - \alpha_t \mathbf{x}_0^1\|^2/(\beta_t^2)} \times 1 \right)}{\left[ \frac{1}{\sum_{i=1}^N e^{\|-\mathbf{x}_t - \alpha_t \mathbf{x}_0^i\|^2/(\beta_t^2)}} \left( e^{-\|\mathbf{x}_t - \alpha_t \mathbf{x}_0^1\|^2/(\beta_t^2)} \times 1 \right) \right]^{w+1}} \\
&= \left[ \frac{\sum_{i=1}^N e^{\|-\mathbf{x}_t - \alpha_t \mathbf{x}_0^i\|^2/(\beta_t^2)}}{e^{\|-\mathbf{x}_t - \alpha_t \mathbf{x}_0^1\|^2/(\beta_t^2)}} \right]^w .
\end{aligned}
\tag{19}
$$

And the radio of two enhanced intermediate distributions is

$$
\frac{Z_t}{Z_0} \left[ \frac{\sum_{i=1}^N e^{\|-\mathbf{x}_t - \alpha_t \mathbf{x}_0^i\|^2/(\beta_t^2)}}{e^{\|-\mathbf{x}_t - \alpha_t \mathbf{x}_0^1\|^2/(\beta_t^2)}} \right]^w .
\tag{20}
$$

Then reconsider the context of Eq. (17), it is observed that the equation does not maintain universal validity. A contradiction between the two sides becomes apparent, as the left side is a function of the stochastic variable $\mathbf{x}_t$, while the right side remains a constant. However, it is easy to check the equation holds when $w = 0$, because when $w = 0$, the left side and the right side of the last line of Eq. (17) are 1. $\qquad\square$

We specifically discuss the case of $q(\mathbf{x}_0|\mathbf{x}_t)$ being a $\delta$ distribution because $q(\mathbf{x}_0|\mathbf{x}_t) = \frac{e^{-\|\mathbf{x}_t - \alpha_t \mathbf{x}_0\|^2}}{\int e^{-\|\mathbf{x}_t - \alpha_t \mathbf{x}_0\|^2} d\mathbf{x}_0}$, which approaches an approximation of a $\delta$ distribution when $t$ is small or when the values of $\mathbf{x}_0$ are highly sparse.

## A.2 PROOF OF COROLLARY 3.1.1

*Proof.* In this case, we have:

$$
\begin{aligned}
\bar{q}(\mathbf{x}_t|\mathbf{c}) &= q(\mathbf{x}_t|\beta, \mathbf{c}) \\
&= q(\mathbf{x}_t|\beta\mathbf{c}).
\end{aligned}
\tag{21}
$$

Treat the $\beta\mathbf{c}$ as an entire $\bar{\mathbf{c}}$, which is a special case of $w = 0$ in Theorem 3.1. $\qquad\square$

## B PROOF OF THEOREM 4.1

*Proof.* Consider the Lagrange form of $R_n(\beta)$:

$$
R_k(\beta) = \frac{\frac{\partial^{k+1} \bar{\epsilon}^\theta (\mathbf{x}_t|\beta\mathbf{c})}{\partial \beta^{k+1}} \Big|_{\beta=\xi}}{(k+1)!} (\beta - 1)^{k+1},
\tag{22}
$$

where $\xi \in [0, \beta]$. Then we can get the upper bound of $\|R_n(\beta)\|$:

$$
\begin{aligned}
\|R_n(\beta)\| &\leq \frac{M_{n+1}}{(n+1)!} (B - 1)^{n+1} \\
&\leq \frac{M_{n+1}}{(n+1)!} B^{n+1}.
\end{aligned}
\tag{23}
$$

$\qquad\square$

To establish a more relaxed condition for the convergence of the sequence $\{M_n \,|\, n \in \mathbb{N}\}$, we utilize Stirling's formula in Eq. (23) to obtain:

$$
\frac{M_{n+1}}{(n+1)!} B^{n+1} \sim \frac{M_{n+1}}{\sqrt{n+1}} \left[ \frac{eB}{n+1} \right]^{n+1},
\tag{24}
$$

which indicates when $n \to +\infty$, the sequence $\{R_n(\beta) \mid n \in \mathbb{N}\}$ converges to 0 if

$$M_{n+1} \sim o\left(\sqrt{n+1}\left[\frac{n+1}{eB}\right]^{n+1}\right).$$

## C  THE UNIQUENESS OF THE SECOND-ORDER TERM

*Proof.* When we use $\frac{y_2 - y_1}{x_2 - x_1}$ of two points $(x_1, y_1), (x_2, y_2)$ to estimate the gradient at $\frac{x_1 + x_2}{2}$, For three data pairs $(x_0, y_0), (x_1, y_1), (x_2, y_2)$, where $(x_1 - x_0)(x_2 - x_1)(x_0 - x_2) \neq 0$, no matter how we organize them, the estimated second-order gradient is uniquely determined as:

$$2\frac{x_0 y_2 + x_1 y_0 + x_2 y_1 - x_0 y_1 - x_1 y_2 - x_2 y_0}{(x_1 - x_0)(x_2 - x_1)(x_0 - x_2)}.$$

Let us define $x_0 = 0$, $x_1 = m$, and $x_2 = 1$. With these values, we can proceed to estimate the second-order gradient, which is given by:

$$\frac{2}{m(1 - m)}\left((1 - m)y_0 + my_2 - y_1\right).$$

$\square$

## D  IMPLEMENTATION DETAILS

### D.1  THE CONDITION SPACE $\mathcal{C}$

In this paper, we have designed two "cone" structures for the conditions of Stable Diffusion. All two kinds $\mathcal{C}$ are extended from the tensors after CLIP model, whose dimension is $77 \times 768$.

- $\mathcal{C}_{\text{all}}$: We use the pretrained CLIP model to extract the text embedding $\mathbf{c}_{\text{text}}$ of the captions. We also get the extract embedding $\mathbf{c}_{\varnothing}$ of empty caption, then with the inner coefficient $\beta$, we get the enhanced embedding $\mathbf{c}_{\varnothing} + \beta(\mathbf{c}_{\text{text}} - \mathbf{c}_{\varnothing})$

- $\mathcal{C}_{\text{nouns}}$: We utilize the pretrained CLIP model to extract the text embedding $\mathbf{c}_{\text{text}}$ from the captions. Additionally, we obtain the embedding $\mathbf{c}_{\varnothing}$ for an empty caption. By incorporating the inner coefficient $\beta$ and the indicator function $\mathbf{1}_{\text{nouns}}$, we can modify the embedding. Specifically, we set the values corresponding to the positions of nouns to 1 in $\mathbf{1}_{\text{nouns}}$, while the remaining values are set to 0. The resulting enhanced embedding is given by $\mathbf{c}_{\varnothing} + \beta\mathbf{1}_{\text{nouns}}(\mathbf{c}_{\text{text}} - \mathbf{c}_{\varnothing})$.

### D.2  DETAILS OF FIN-TUNING PROCESS

We set $rank = 4$ and apply the Low-Rank Adaptation (Hu et al., 2022; Ruiz et al., 2023) to modify the attention layers of the U-Net (Ronneberger et al., 2015) of Stable Diffusion v1.5 (Rombach et al., 2022). We use the Adam optimizer with a learning rate of $1e - 4$ and a batch size of 8. We fine-tune the model for 300 epochs on a small part of MS-COCO (Lin et al., 2014) dataset, which contains 30 images and their corresponding captions. We use the pretrained Stable Diffusion v1.5 model as the initialization of the U-Net. We compare our fine-tuning policy with default fine-tuning policy.

## E  SAMPLES AFTER FINE-TUNING

After the fine-tuning process, we proceed to compare the samples generated using different training policies. The corresponding results are presented in Figure 3 and Figure 4. Each sample is generated with the caption "a brown and white giraffe in a field of grass" and arranged from left to right, with values of inner $\beta$ set to 1.0, 1.2, 1.4, and 1.6. Notably, our training policy demonstrates superior performance compared to the default training policy when $\beta$ assumes relatively larger values. This observation suggests that our training policy effectively captures the inherent "cone" structure of $\mathcal{C}$.

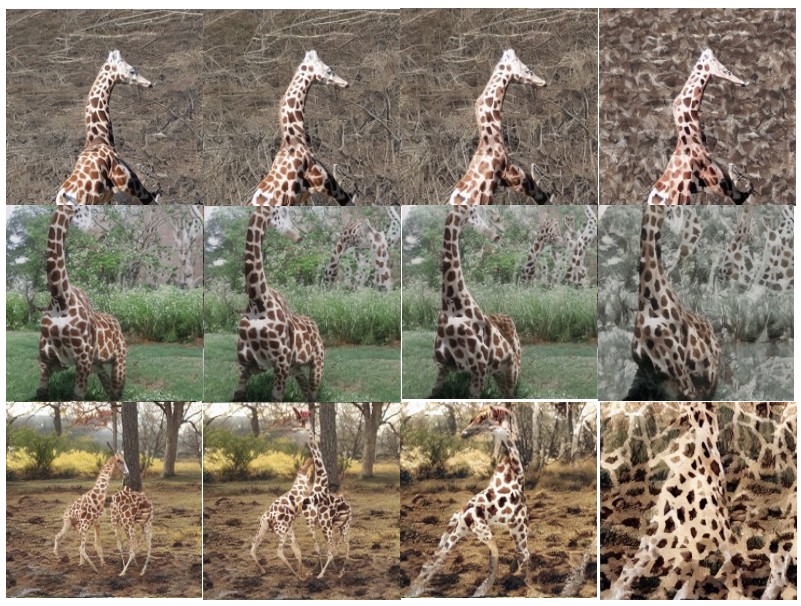

Figure 3: Generated images of different inner $\beta$ of our training policy.

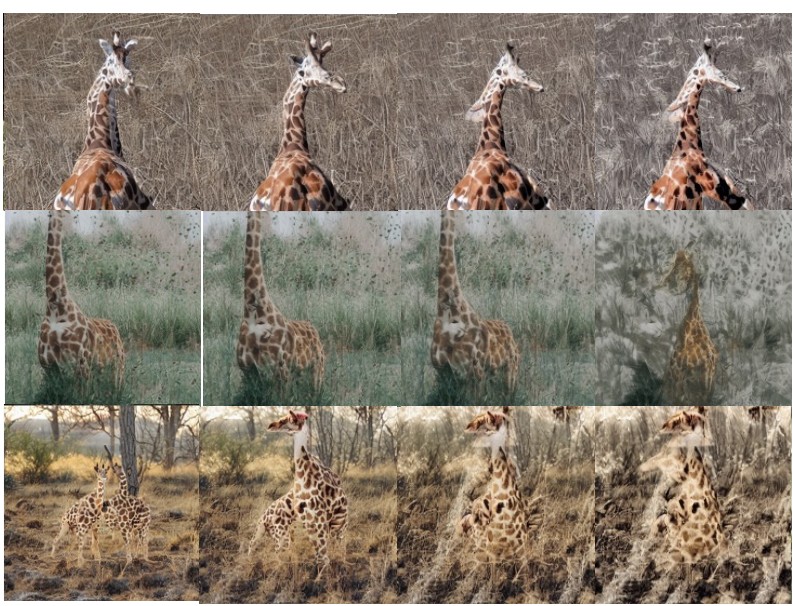

Figure 4: Generated images of different inner $\beta$ of default training policy.

Table 4: FID results on U-ViT of CFG and ICFG.

| Steps | 5w | 10w | 15w | 20w | 25w | 30w | 35w | 40w | 45w | 50w | 55w | 60w | 65w | 70w | 75w | 80w |
|---|---|---|---|---|---|---|---|---|---|---|---|---|---|---|---|---|
| CFG | 34.23 | 13.64 | 11.26 | 10.38 | 9.78 | 8.98 | 8.98 | 8.76 | 8.58 | 8.52 | 8.37 | 8.37 | 8.27 | 8.32 | 8.39 | 8.10 |
| ICFG | 24.69 | 13.51 | 11.00 | 10.13 | 9.69 | 9.09 | 8.82 | 8.68 | 8.54 | 8.41 | 8.35 | 8.21 | 8.29 | 8.15 | 8.10 | **7.92** |

Table 5: Experiments about the speedup.

| Method | Time (seconds) | Extra Time | U-Net Computation | FID | CLIP Score |
|---|---|---|---|---|---|
| CFG | $10.43 \pm 0.23$ | 0% | 100% | 15.42 | 25.80 |
| 2nd-order ICFG | $15.01 \pm 0.31$ | 43.91% | 150% | 15.28 | 26.11 |
| 0.2-0.8 2nd-order ICFG | $13.17 \pm 0.26$ | 26.27% | 130% | 15.29 | 26.03 |

## F  VISUAL RESULTS COMPARISON

We conducted a comparison between CFG and second-order ICFG with $w = 5.0$ and $v = 0.25$. The visual comparisons are presented in Figure 5. It is evident from the images that second-order ICFG outperforms CFG, producing images with better-rendered hands and closer alignment to the provided prompts.

## G  EXPERIMENTS ON ANOTHER FRAMEWORK

we train another framework, U-ViT (Bao et al., 2023a), with a resolution of 256x256 on the COCO dataset from scratch to fully explore the capabilities of our ICFG. The FID results are listed in Table 4.

## H  EXPERIMENTS ABOUT THE SPEEDUP

We conducted our experiments on an NVIDIA GeForce RTX 3090, using a batch size of 4. We performed 100 samplings to calculate the timings and utilized 10,000 images for the computation of FID and CLIP scores, The results are shown in Table 5.

We have the following findings.

- Due to the text encoder and VAE decoder, the real-time consumption of the 2nd-order ICFG is less than the estimated extra computation of the U-Net.
- Through a preliminary selection of key timesteps (0.2-0.8) for applying the 2nd-order ICFG, we achieve nearly full FID benefits and a 74% improvement in CLIP scores, with a reduced extra inference time of 26.27%. We anticipate further enhancements in extra inference time by refining the selection of key timesteps.

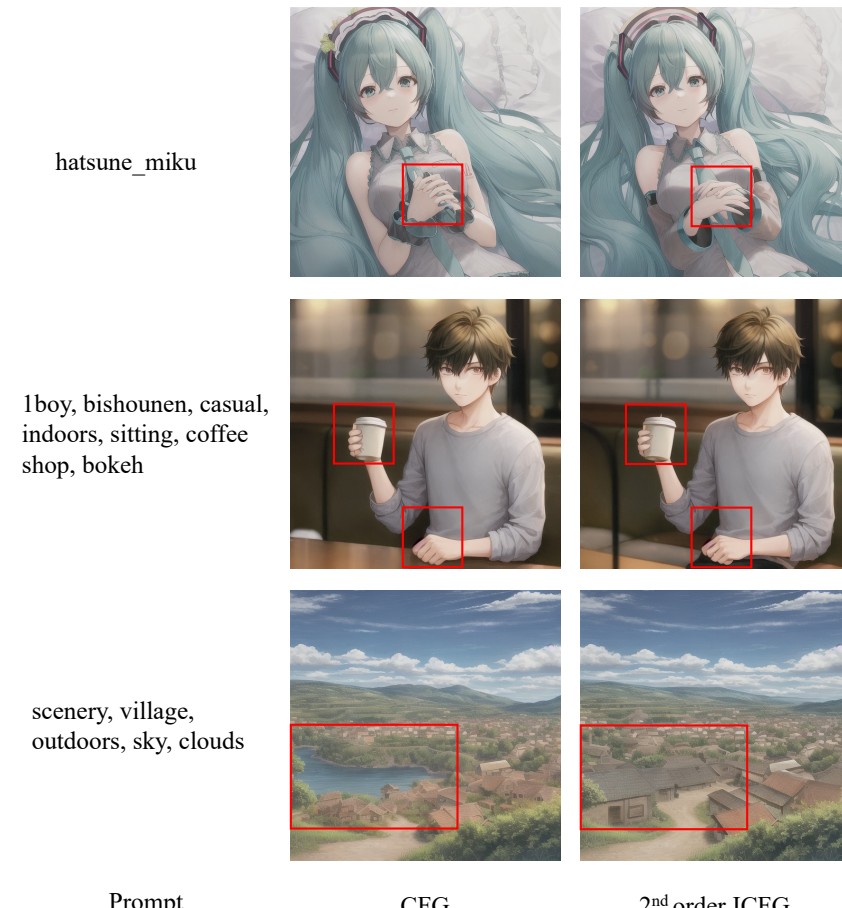

Figure 5: The generated images presented here compare the outputs of CFG and second-order ICFG with $w = 5.0$ and $v = 0.25$, utilizing the model anything-v4.0 (https://huggingface.co/xyn-ai/anything-v4.0). In the first two rows, it is evident that our second-order ICFG produces superior results in hand generation. In the last row, our second-order ICFG generates images that align more closely with the provided prompts.

