# OpenReview forum: "Inner Classifier-Free Guidance and Its Taylor Expansion for Diffusion Models"
_ICLR.cc/2024/Conference — ICLR 2024 poster_

### Official Review · Reviewer_UdAr · 2023-10-31

**Soundness:** 1 poor
**Presentation:** 2 fair
**Contribution:** 2 fair
**Rating:** 5
**Confidence:** 4

**Summary:**

This paper introduces an enhancement of classifier-free guidance (CFG) for diffusion models, called inner classifier-free guidance (ICFG). The paper claims that CFG can be extended to ICFG when the condition is continuous, leading to further improvements. They provide a theoretical analysis based on the condition space assumption and Taylor expansion. They present the experimental results comparing CFG and ICFG.

**Strengths:**

The idea of combining the guidance strength and condition is interesting because adjusting guidance strength is the effect of a trade-off between image fidelity and condition information.

**Weaknesses:**

1. Concerns about the theoretical analysis in Section 3
* In the proof of Theorem 3.1, there is doubt about the validity of the third equality in Eq. 16. It seems that it may come from Eq. 15, which might change the equality to the proportion.
* Additionally, the relationship between the last two terms in Eq. 16 is the proportional relationship, but I'm not sure why these two terms are equal after the logarithm, as shown in Eq. 17.
* In the proof of Theorem 3.1, it would be helpful to provide a detailed explanation of why there is a contradiction unless $w=0$ or $q(x_0|x_t)$ is a Dirac delta distribution.
* Theorem 3.1 claims necessary and sufficient conditions, but the proof only demonstrates one direction.
* In the paper, all cases of the enhanced intermediate distribution are denoted $\bar{q}$. (In addition, $\beta(x_t)$ in Eq. (12) is not defined.) This is confusing, especially for the definition $\bar{q}(x_t|c,\beta):=\bar{q}(x_t|\beta c)$.
* In the proof of Corollary 3.1.1, it is unclear whether the first proportion holds. It seems that $q$ is modified by $\bar{q}$. Detailed derivation and explanation would be helpful.

2. Not significant experimental results
* The quantitative results suggest that the performance gain is marginal, and this may require very careful hyperparameter tuning. It would be beneficial to include results from other datasets to assess the tuning problem.
* The implementation of ICFG in Algorithms 2 and 3 requires three network evaluations for each timestep ($\epsilon(z_i), \epsilon(z_i,c), \epsilon(z_i,mc)$), which is 1.5 times more network evaluations than CFG. Consequently, ICFG has 1.5 times higher sampling cost compared to CFG.

**Questions:**

Please answer the questions in the Weaknesses section.

---

> ### Author Response · Authors · 2023-11-20
> **Rebuttal part I**
>
> Our gratitude extends to Reviewer UdAr for acknowledging the novelty in our approach to combining guidance strength and condition. Their assistance in correcting definitions, perfecting proofs and the formulation of the theorem, along with their guidance in including additional empirical results and speeding up our method, has been invaluable.
>
> ### **Q1: Concerns about the theoretical analysis in Section 3.**
>
> Thank you very much for your valuable suggestions. We acknowledge that our theoretical analysis had some shortcomings. To address these issues, we have made several improvements in the proof, and we would like to highlight the key changes:
>
> - We have explicitly written out the normalization constant.
> - We provide a clearer formula for typical training scenarios.
> - We have revised Theorem 3.1, retaining only the sufficient conditions and acknowledging that we will prove their necessity for typical training scenarios later.
>
> We apologize for any confusion caused by the symbol system. The enhanced intermediate distribution is consistently denoted as $\bar{q}$, but when explicitly expressing the enhancement strength $\beta$, we can omit the symbol $\bar{q}$. The correct formula is as follows:
>
> $$
> \bar{q}(x_t|c) = q(x_t | c, \beta)=q(x_t|\beta c)
> $$
>
> This equation holds under the cone assumption, allowing the neural network to capture them together. The Corollary 3.1.1 is now more clear in this context.
>
> We also apologize for the lack of interpretation. In Eq. 12, the term $\beta(x_t)$ is a general energy function, chosen arbitrarily to modify the enhanced intermediate distribution, and it is unrelated to the guidance strength $\beta$.
>
> Thank you once again for your insightful questions. We believe that these improvements significantly enhance the theoretical foundation of the manuscript.
>
> ### **Q2-1: The improvement of second-order ICFG is not clear.**
>
> We apologize for any confusion in our experiment descriptions. While maintaining **the same training policy as CFG**, the continuous feature of the condition space enables second-order ICFG to **consistently outperform** CFG in both FID and CLIP scores across various CFG strengths ($w$).
>
> For a clearer understanding of our advancements in generation quality, consider the comparison between second-order ICFG with $w = 2.0, v = 0.25, C=C_{all}$ and CFG with $w = 3.0$ (depicted as the second orange spot and the third blue spot in Figure 2). Despite nearly identical CLIP scores (26.12 and 26.11), **we observed a notable improvement of $(16.68 - 15.28)/16.68 =$ 8.3% in FID**. This implies that with increased control strength, the generated quality is significantly enhanced.
>
> It is noteworthy that the **CLIP Score improvement achieved by our second-order ICFG is nearly half of the overall improvements observed from Stable Diffusion v1.1 to v1.5** ([comparison chart](https://huggingface.co/CompVis/stable-diffusion/resolve/main/v1-1-to-v1-5.png)). This indicates a substantial improvement in performance.
>
> Due to GPU limitations, we were unable to train SD from scratch. However, we are currently training another framework, U-ViT, with a resolution of 256x256 on the COCO dataset from scratch to fully explore the capabilities of our ICFG. The preliminary results are listed below, and we plan to update them upon completion of the training.
>
> | Training Policy \FID \ Training Steps| 5w | 10w | 15w | 20w | 25w | 30w | 35w | 40w | 45w | 50w| 55w | 60w | 65w | 70w | 75w | 80w |
> |----------|:-------------:|------:|--:|--:|--:|--:|--:|--:|--:|--:|--:|--:|--:|--:|--:|--:|
> | Original | 34.23 | 13.64 | 11.26 | 10.38 | 9.78 | 8.98 | 8.98 | 8.76 | 8.58 | 8.52 | 8.37 | 8.37 | 8.27 | 8.32 | 8.39 | 8.10 |
> | ICFG | 24.69 | 13.51 | 11.00 | 10.13 | 9.69 | 9.09 | 8.82 | 8.68 | 8.54 | 8.41 | 8.35 | 8.21 | 8.29 | 8.15 | 8.10 | **7.92** |
>
> We plan to report the final FID and CLIP scores after completing the entire training process. As of now, we have observed a notable **3% improvement** in FID with  **ICFG training algorithm and CFG sampling algorithm** compared with original training policy.
>
> ### **Q2-2: Other datasets and carefully hyperparameter tuning.**
>
> Thank you very much for the suggestions. We will consider use more datasets to tune the $v$ and middle point in a more data driven way.
> In fact we just choose the $v$ and middle point under the following consideration:
> 1. Because the U-Net doesn't trained to realize the cone structure, **middle point should be near the original condition point.**
> 2. Due to  the reminder term $R_2{(\beta)}$ will be larger with larger $\beta$ (Because $R_n{(\beta)}\sim o((\beta-1)^n)$), **$\sqrt{v}$ is an upper bound of the 2nd order "step size" and should be small**.

---

> ### Author Response · Authors · 2023-11-20
> **Rebuttal part II**
>
> ### **Q3: Extra computation cost.**
>
> In comparison, CFG requires 2 forward passes, meaning our method incurs a 50% increase in computation. To further minimize this additional computation, we propose a method to selectively apply 2nd order ICFG only during key timesteps. Upon examining the insights from the second figure in Figure 1, it is apparent that the guidance is more significant during the middle approximately 40% of the time. **Capitalizing on this observation, our plan is to implement the second ICFG exclusively during these specific time steps.** This strategic approach is expected to result in a substantial reduction, decreasing the extra inference time from 50% to approximately 20%.

---

> > ### Comment · Reviewer_UdAr · 2023-11-22
> >
> > Thank you for the response. The author's response addressed some of my concerns, but I have the following additional concerns.
> >
> > Q1. I wonder if there is any problem in this study by reducing the theorem to the sufficient condition of the original Theorem 3.1
> >
> > Q2. So far, they may not provide any experimental results on other datasets, so I think it is hard to confidently claim that there is a performance improvement. Specifically, I think that the consideration of hyperparameter tuning needs to be claimed with experimental evidence.
> >
> > Q3. As another reviewer pointed out, the proposed method that improves the computational cost requires experimental results to show that the performance is guaranteed.

---

> ### Author Response · Authors · 2023-11-22
>
> Thank you sincerely for your prompt and valuable feedback. We also acknowledge and apologize for any confusion resulting from our previous incomplete statement in the rebuttal.
>
> ### Q1: If there is any problem in this study by reducing the theorem to the sufficient condition of the original Theorem 3.1.
>
> There is no issue since the sufficient condition is enough to demonstrate that our ICFG does not exhibit inconsistency in the enhanced intermediate distributions.
>
> ### Q2: Experiments on other datasets.
>
> Due to limitations of time, we haven't extensively tuned hyperparameters on larger datasets like Laion. However, to **demonstrate the stability and universal applicability of the existing improvements**, we have taken the following two measures:
>
> 1. For FID and CLIP Score, we conducted five calculations to present mean and standard deviation results for key hyperparameters, specifically setting $middle = 1.1$ and $v = 0.25$. Although we have completed only three points so far, we plan to extend this analysis further. Analyzing our existing experimental results, it's noteworthy that **the standard deviation is significantly smaller than the observed improvement.** This pattern suggests a high degree of stability in our achieved enhancements.
>
> CLIP Score (higher values indicate better performance):
>
> |  $w$  |  1 | 2|  3      |   4 | 5|
> |---- |:------:|:------:|:------:|:------:|:------:|
> |CFG|25.03|25.80|26.12|26.34|26.45|
> |2nd-order ICFG| **25.46** | $26.13\pm0.02$ |$26.36\pm0.02$| $26.51\pm0.03$ | **26.59**|
>
> FID (lower values indicate better performance):
>
> |  $w$  |  1 | 2|  3      |   4 | 5|
> |---- |:------:|:------:|:------:|:------:|:------:|
> |CFG| 17.24 | 15.42 | 16.68 | 18.18 | 19.53|
> |2nd-order ICFG|**16.40**|**$15.25\pm 0.03$**|$16.58\pm 0.04$|$17.97 \pm 0.03$|**19.35**|
>
> 2. We have also included several visual comparisons on anything-V4 ([https://huggingface.co/xyn-ai/anything-v4.0](https://huggingface.co/xyn-ai/anything-v4.0)). It is evident from the images that second-order ICFG outperforms CFG, **producing images with better-rendered hands and closer alignment to the provided prompts**. **The results are visualized in Figure 5 of Appendix F.** It shows the universal applicability of our ICFG.
>
> We are downloading the Laion dataset and we will further tune the hyperparameters for the future work.
>
> ### Q3: Experiments about the speedup.
>
> We conducted our experiments on an NVIDIA GeForce RTX 3090, using a batch size of 4. We performed 100 samplings to calculate the timings and utilized 10,000 images for the computation of FID and CLIP scores.
>
> |  Method  |   Time (seconds)| Extra Time Compared with CFG|  Estimation of U-Net Computation      |   FID |CLIP Score |
> |----------|:-------------:|:------:|:------:|:------:|:------:|
> | CFG |10.43 $\pm$ 0.23|0%| **100%** |    15.42 | 25.80 |
> | 2nd-order ICFG |15.01 $\pm$ 0.31|43.91%|  150% | **15.28** |**26.11** |
> | 0.2-0.8 2nd-order ICFG |13.17 $\pm$ 0.26| 26.27%|**130%** |    **15.29** |**26.03** |
>
> * Due to the text encoder and VAE decoder, the real-time consumption of the 2nd-order ICFG is less than the estimated extra computation of the U-Net.
> * Through a preliminary selection of key timesteps (0.2-0.8) for applying the 2nd-order ICFG, we achieve nearly **full FID benefits and a 74% improvement in CLIP scores**, with a **reduced extra inference time of 26.27%**. We anticipate further enhancements in extra inference time by refining the selection of key timesteps.

---

> > ### Comment · Reviewer_UdAr · 2023-11-22
> >
> > Thank you for your prompt response. In particular, it is great that they provided experiments on the computational cost of ICFG. Therefore, I would change the score from *reject* to *borderline reject*. The reason I am still on the reject side is that additional experimental results on other datasets are missing anyway, and the theoretical results were done during the rebuttal period and need more consideration. I will consider the author's response and have further discussions with the other reviewers.

---

> > > ### Author Response · Authors · 2023-11-23
> > >
> > > Dear Reviewer UdAr,
> > >
> > > As the discussion stage approaches its conclusion, we would like to inquire if you have any additional questions or suggestions. We would be delighted to engage in further discussion with you.

---

> > > > ### Comment · Reviewer_UdAr · 2023-11-23
> > > >
> > > > Thank you for the further explanation. For now, I will keep my rating, but I will discuss it with other reviewers.

---

> > > > > ### Author Response · Authors · 2023-11-23
> > > > >
> > > > > We extend our sincere gratitude to you once again for your invaluable reviews and suggestions.
> > > > >
> > > > > Happy Thanksgiving!
> > > > >
> > > > > Warm wishes from all authors.

---

> ### Author Response · Authors · 2023-11-23
>
> Dear reviewer UdAr,
>
> Thank you sincerely for appreciating our previous rebuttal. We understand your concerns, and to further alleviate them, we have included additional information **regarding the results on other datasets**. We also provide more information to make it easier for you to review and validate our proof.
>
> ### Experiments on other datasets.
>
> 1. We want to clarify the previous response to Q2. The visual comparisons in Figure 5 of Appendix F were conducted on the anything-V4 dataset ([https://huggingface.co/xyn-ai/anything-v4.0](https://huggingface.co/xyn-ai/anything-v4.0)) **with the same hyperparameters $middle = 1.1$ and $v = 0.25$**. It is important to note that **this model was trained on other datasets**, specifically in the anime domain.
> 2. Additionally, to address the concerns raised by reviewer UdAr, we performed CLIP Score comparisons **on the pokemon-blip-captions dataset** ([https://huggingface.co/datasets/lambdalabs/pokemon-blip-captions](https://huggingface.co/datasets/lambdalabs/pokemon-blip-captions)). The results involve sampling algorithms ICFG and CFG, along with both Stable Diffusion v1.5 and U-Net fine-tuned models. We used the script available at [https://github.com/huggingface/diffusers/blob/main/examples/text_to_image/train_text_to_image_lora.py](https://github.com/huggingface/diffusers/blob/main/examples/text_to_image/train_text_to_image_lora.py). The reported CLIP Score varies with different $v$. It's important to note that FID is not reported due to the dataset's small size, making it impractical to calculate FID with the most general settings (as referenced in [https://github.com/bioinf-jku/TTUR/issues/4](https://github.com/bioinf-jku/TTUR/issues/4)).
>
> For the original Stable Diffusion v1.5, with  $w = 2.0$:
>
> |  $v$  |  0| 0.25|  0.5      |   0.75 | 1.0|
> |---- |:------:|:------:|:------:|:------:|:------:|
> |CLIP Score|8.38|8.40|8.41|8.42|**8.44**|
>
> For the fine-tuned Stable Diffusion v1.5 on the pokemon-blip-captions dataset, with $w=2.0$:
>
> |  $v$  |  0| 0.25|  0.5      |   0.75 | 1.0|
> |---- |:------:|:------:|:------:|:------:|:------:|
> |CLIP Score|13.70|13.75|13.77|13.79|**13.80**|
>
> Despite the **untuned middle points and consequently suboptimal results**, **2nd-order ICFG consistently outperforms CFG**. Notably, with larger $v$ in the range from 0 to 1, it exhibits superior CLIP Score performance, **aligning with the findings in COCO**. We consider to incorporate additional results from larger datasets in the final version.
>
> ### Concern on the proof.
>
> We appreciate your concern, and we would like to highlight key points for validation of the proof.
>
> * When $w \ne 0$, there exists an inconsistency between the enhanced intermediate distributions. One approach involves obtaining the conditional $x_0$ and then disturbing it, while the other involves acquiring the disturbed $x_t$ and subsequently applying the condition.
> * However, when $w = 0$, there is no such inconsistency.
> * Due to our approach of placing the condition enhancement in the condition space, our ICFG consistently exhibits $w=0$ in the score domain, effectively circumventing any potential inconsistency.
>
> We extend our sincere gratitude to you once again.

---

### Official Review · Reviewer_jYKn · 2023-11-01

**Soundness:** 2 fair
**Presentation:** 2 fair
**Contribution:** 2 fair
**Rating:** 6
**Confidence:** 4

**Summary:**

This paper presents a generalized version of classifier-free guidance (CFG), i.e., inner classifier-free guidance. By exploiting the continuity of the generation condition, the author propose an interesting taylor expansion formulation to interpret CFG, where the classic CFG is viewed as the first order case of the proposed formulation. Given such novel formulation, the author proposed higher order version of CFG to achieve better image generation results.

**Strengths:**

1. The proposed formulation is novel and insightful.
2. The paper is easy-to-follow.
3. The hyper-parameters introduced by the method is well-studied.

**Weaknesses:**

1) In the reviewer's viewpoint, the major weakness of the current submission is that the empirical validation is not sufficient. More qualitative results should be provided to justify the effectiveness of the method. One example is that it would be better if the provided qualitative samples could corroborate with the numerical experimental results. Another example is that the author could provided some examples showing that ICFG can resolve some of the well-known failure cases of Stable Diffusion.

2) In addition, it would be good to show the effectiveness of ICFG on other fine-tuned variation of Stable Diffusion such as anything-V4 (https://huggingface.co/xyn-ai/anything-v4.0), etc.

**Questions:**

1. Is the few-shot training of ICFG on a small dataset generalizable to large scale setting? For example, as I understand, in this work, the authors trained with the ICFG policy on COCO dataset. I was wondering if this trained LoRA is readily available for generation beyond COCO dataset (e.g., on laion-level dataset)?

2. Is the proposed ICFG applicable on other fine-tuned variation of Stable Diffusion such as anything-V4 (https://huggingface.co/xyn-ai/anything-v4.0), etc.

---

> ### Author Response · Authors · 2023-11-20
> **Rebuttal by Authors**
>
> We are deeply thankful for Reviewer jYKn's recognition of our novel formulation and well-studied hyper-parameters. We sincerely value their guidance in recommending the inclusion of additional empirical results, particularly in the form of visual comparisons.
>
> ### **Q1: Need more experiment results.**
>
> We added more experiments here, and after discussion, we will add all the experiments to the final **manuscript**.
>
> 1. For the training algorithm, due to GPU limitations, we were unable to train SD from scratch. However, we are currently training another framework, U-ViT, with a resolution of 256x256 on the COCO dataset from scratch to fully explore the capabilities of our ICFG. The preliminary results are listed below, and we plan to update them upon completion of the training.
>
> | Training Policy \FID \ Training Steps| 5w | 10w | 15w | 20w | 25w | 30w | 35w | 40w | 45w | 50w| 55w | 60w | 65w | 70w | 75w | 80w |
> |----------|:-------------:|------:|--:|--:|--:|--:|--:|--:|--:|--:|--:|--:|--:|--:|--:|--:|
> | Original | 34.23 | 13.64 | 11.26 | 10.38 | 9.78 | 8.98 | 8.98 | 8.76 | 8.58 | 8.52 | 8.37 | 8.37 | 8.27 | 8.32 | 8.39 | 8.10 |
> | ICFG | 24.69 | 13.51 | 11.00 | 10.13 | 9.69 | 9.09 | 8.82 | 8.68 | 8.54 | 8.41 | 8.35 | 8.21 | 8.29 | 8.15 | 8.10 | **7.92** |
>
> We plan to report the final FID and CLIP scores after completing the entire training process. As of now, we have observed a notable **3% improvement** in FID with  **ICFG training algorithm and CFG sampling algorithm** compared with original training policy.
>
> 2. For the sampling algorithm, We incorporated CLIP Scores on the fine-tuned anything-v4.0 using prompts from the test dataset available at a [prompt dataset](https://huggingface.co/datasets/Gustavosta/Stable-Diffusion-Prompts/tree/main/data).
>
> |  Method  |      w=2.0,v=0;    |  w=2.0, v=0.25; | w=6.0, v = 0;| w = 6.0, v=0.25 |
> |----------|:-------------:|------:|------:|------:|
> | CLIP Score |  11.77 |    **11.81** | 10.72 |10.73 |
>
> We observe that our sampling algorithm **is applicable to other fine-tuned Stable Diffusion** (SD)-based models.
>
> 3. We have also included several visual comparisons on anything-V4 ([https://huggingface.co/xyn-ai/anything-v4.0](https://huggingface.co/xyn-ai/anything-v4.0)). It is evident from the images that second-order ICFG outperforms CFG, producing images with better-rendered hands and closer alignment to the provided prompts. **The results are visualized in Figure 5 of Appendix F.**
>
> Furthermore, we plan to include more additional qualitative samples, contemplating the addition of more visual comparisons to vividly demonstrate the effectiveness of our methods.
>
> ### **Q2: Is the few-shot training of ICFG on a small dataset generalizable to large-scale setting?**
>
> We apologize for any confusion. For the training algorithm, we encourage the application of the training algorithm to the entire dataset to more comprehensively capture the guidance strength in the U-Net during training.
> However, **the primary improvements highlighted in Table 1 do not require fine-tuning**; the ICFG training algorithm is not necessary. We believe that the effectiveness of the second-order term is attributed to the continuity of the condition space. Consequently, **we consider the improvements demonstrated by the second-order ICFG to be generalizable to large-scale settings**.
>
>
> ### **Q3: It would be good to show the effectiveness of ICFG on other fine-tuned variations of Stable Diffusion such as anything-V4 ([https://huggingface.co/xyn-ai/anything-v4.0](https://huggingface.co/xyn-ai/anything-v4.0)), etc.**
>
> As explained in Q1, the second-order ICFG can be applied immediately and gain improvements. Furthermore, we plan to include more additional qualitative samples, contemplating the addition of more visual comparisons to vividly demonstrate the effectiveness of our methods.
>
> As explained in response to Q1, the second-order ICFG can be applied immediately to achieve improvements. We have included several visual comparisons on anything-V4 ([https://huggingface.co/xyn-ai/anything-v4.0](https://huggingface.co/xyn-ai/anything-v4.0)). Additionally, we are in the process of including more qualitative samples. We are considering the addition of more visual comparisons to vividly demonstrate the effectiveness of our methods.

---

> ### Author Response · Authors · 2023-11-23
>
> Dear reviewer jYKn,
>
> Thank you for your invaluable review. We have thoroughly considered your concerns, particularly those pertaining to **further qualitative sample comparison experiments of fine-tuned variations of Stable Diffusion**, and the universal applicability of our training and sampling algorithm. In our response, we have diligently addressed these issues to the best of our ability.
>
> As the discussion stage approaches its conclusion, we would like to inquire if you have any additional questions or suggestions. We would be delighted to engage in further discussion with you.

---

> > ### Comment · Reviewer_jYKn · 2023-11-23
> >
> > Thanks for the reply. In light of these additional results, I raised my score to 6. Please consider adding these results in the updated version.

---

> > > ### Author Response · Authors · 2023-11-23
> > >
> > > Thanks you very much for your invaluable review, we will add all the results of the discussion in the final version.

---

### Official Review · Reviewer_LfUU · 2023-11-07

**Soundness:** 4 excellent
**Presentation:** 4 excellent
**Contribution:** 3 good
**Rating:** 8
**Confidence:** 3

**Summary:**

This paper generalizes CFG for diffusion model guidance by adapting the guidance strength according to the relevance between the condition and a given sample. Conditions more relevant to a sample require weaker guidance strength.

**Strengths:**

The paper identifies an issue with the deviating SDE when guidance is added and proposes a solution with clearly defined assumptions. The paper strikes a good balance between theoretical analysis and empirical results. The theories are relevant to the technique and justify the design choice. Extensive ablation studies on hyperparameters of the method yield insight to the adoption of the technique in practice.

**Weaknesses:**

**Cone assumption**

It seems like the key point of ICFG working is for the conditional space to be a cone. Is there any method to check whether a conditional space is a cone beforehand for practioners to decide whether ICFG should be adopted? Are there any metrics for characterizing how cone-shaped a conditional space is?

**Algorithm 3 relaxing 2nd order term**

The relaxation of the 2nd order term is concerning. The argument for ICFG with 2nd order Tayler Expansion working better than typical CFG is the additional information provided by the 2nd order term. However, if an additional hyperparmeter is introduced and optimized over, does this argument still hold? Does the performance gain of ICFG truly come from the 2nd order information or is it the mere additional of another tuning knob $v$?

**3 forward passes for naive 2nd order ICFG implementation**

The last paragraph of the discussion section mentions one major drawback of 2nd-order ICFG, which would require 3 forward passes to estimate the 2nd order term. The authors mention a solution of "reusing the points for the pre-order term". Elaboration on how exactly this can be done is crucial for actual adoption of this technique. Paying a computation penalty of 3 forward passes is definitely not feasible. The authors should also compare theoretically and/or empirically how this approximation would affect the efficacy of ICFG.

**Questions:**

(see weaknesses)

Willing to increase score if issues are adequately handled.

---

> ### Author Response · Authors · 2023-11-20
> **Rebuttal by Authors**
>
> We extend our appreciation to Reviewer LfUU for acknowledging both our theoretical analysis and empirical results. Their guidance in accelerating our method and perfecting the non-strict 2nd order term reasons has been tremendously helpful.
>
> ### **Q1: Cone assumption.**
>
> We appreciate your understanding, and we would like to clarify Assumption 3.1 for better comprehension. In practice, we utilize the CLIP embedding space as the condition space, which is 77x768-dimensional. To construct a cone structure, we use a finite number of outputs from CLIP as the basis, extending it by multiplying it with any positive number.
>
> The true assumption, in essence, is that there are no identical directions among the outputs of CLIP for different texts,  which is more realistic. Importantly, Assumption 3.1 aligns with CFG. While CFG operates in the score domain (output of U-Net), ICFG applies the cone assumption within the condition space.
>
> The challenge posed by Assumption 3.1 lies in adapting the score predictor (U-Net for Stable Diffusion) because, during training, the U-Net has never encountered the extended cone structure. However, owing to the continuity of the extended condition space, the unadapted U-Net performs well within a small range of the cone structure. Consequently, the approximated second-order ICFG with the unadapted U-Net still demonstrates a notable improvement compared to CFG.
>
> In summary, the cone is typically constructed by extending the original conditional space. The crucial factor is whether the score predictor can adapt to it. If the space is continuous, it usually works within a small range. **From our perspective, if the space can be extended to form a cone, the ICFG should be adaptable within a limited range.**
>
> **To assess the U-Net's ability to capture the cone structure, we employ two complementary approaches.** Firstly, we directly apply the guidance strength on the condition space to observe the generated results, as demonstrated in Appendix E. Secondly, we calculate the FID and CLIP Score of the samples as metrics for evaluation.
>
>
>
> ### **Q2: Algorithm 3 relaxing 2nd order term.**
>
> Thank you very much for your question, which prompted significant revisions to our manuscript. In the introduction of the non-strict approach, the parameter $v$ does make logical sense.
>
> Given the order of approximation, the remainder term $R_n{(\beta)}$ becomes larger with increasing $\beta$ (since $R_n{(\beta)}\sim o((\beta-1)^n)$), leading to potentially worse results. Therefore, a strict 2nd-order algorithm is more effective for smaller $\beta$. Additionally, we considered whether limiting $(\beta - 1)^2$ with an upper bound $v$ would result in consistent improvements for larger $\beta$ values.
>
> Experiments confirmed our hypotheses. Viewing $v$ as a tuning parameter is also meaningful: $\sqrt{v}$ serves as an upper bound for the 2nd order "step size." Therefore, from an experimental standpoint, the non-strict 2nd-order algorithm appears to be more suitable.
>
>
> ### **Q3: Forward passes for naive 2nd order ICFG implementation.**
>
> The calculation is a crucial aspect of our concern. The nth-order term itself requires n+1 points for estimation. In this computation, we need $\frac{(n+1)(n+3)}{2}$ forward passes to obtain all the terms. However, the nth order term can reuse n points to estimate the (n-1)th order term, reducing the actual forward passes to 3 for 2nd order ICFG.
>
> In comparison, CFG requires 2 forward passes, meaning our method incurs only a 50% increase in computation. To further minimize this additional computation, we propose a method to selectively apply 2nd order ICFG only during key timesteps. Upon examining the insights from the second figure in Figure 1, it is apparent that the guidance is more significant during the middle approximately 40% of the time. **Capitalizing on this observation, our plan is to implement the second ICFG exclusively during these specific time steps.** This strategic approach is expected to result in a substantial reduction, decreasing the extra inference time from 50% to approximately 20%.

---

> > ### Comment · Reviewer_LfUU · 2023-11-22
> >
> > The authors' response is detailed and clarified some issues. Here are some follow ups:
> >
> > Q1: The specific instantiation on the CLIP embedding space illustrates the cone shape assumption well. Please consider adding the concrete example early in the paper (e.g. in page 1 last paragraph).
> >
> > Q2: I don't believe the question is answered so allow me to elaborate. When BatchNorm was first introduced, their authors attributed its success to mitigating internal covariate shift (ICS). However, the newly introduced parameters to model the affine transform suggests that ICS is not the reason of success. In this paper, it is claimed that the second order term provides important information but also introduced Algo 3 with a hyperparameter (that requires tuning) to scale the second order term. This suggests that in general adding the second order term likely doesn't benefit the approximation. Perhaps reporting the experiment results of sweeping for a range of $v$ would convince readers that the hyperparameter is not sensitive.
> >
> > Q3: The explanation of why the naive implementation of ICFG is 1.5x computation of CFG is quite clear. However, the implicit computation cost includes tuning of the hyperparameter $v$ is hidden. Also, the suggested implementation of performing ICFG only in key timesteps requires testing for its empirical efficacy. The estimation of speedup is currently not supported by any experiments.
> >
> > Overall, the insights from theory don't necessarily translate to empirical design choices and that would indicate that the theory, as elegant as it might seem, isn't exactly useful.

---

> > > ### Author Response · Authors · 2023-11-22
> > >
> > > Thank you very much for your prompt and valuable feedback, and we apologize for the confusion caused by our previous statement in the rebuttal.
> > >
> > > ### Q1:  Consider adding the concrete example early in the paper (e.g. in page 1 last paragraph).
> > >
> > >
> > > We genuinely appreciate your feedback, and we are committed to incorporating it into the final version of our manuscript. Specifically, we will include the concrete example in the last paragraph of page 1. Thank you sincerely for your valuable contribution to enhancing the readability of our manuscript.
> > >
> > > ### Q2-1: Confused explanation of original A2.
> > >
> > > We apologize for any confusion caused by our previous explanation and hope that this clarification provides more helpful information.
> > >
> > > The correction term introduced in our theory is given by $\sum_{k = 2}^{+\infty} \frac{1}{k!} \frac{\partial_{\beta = 1}^k \overline{\mathbf{\epsilon}}^\theta(\mathbf{x}_t | \beta \mathbf{c})}{\partial \beta^k}\bigg| (\beta - 1)^k$ (**including infinite terms**), where $\beta = w + 1$. Notably, **when $w$ is small, the second-order term dominates**.
> > >
> > > However, **when $w$ is relatively large**, the second-order term may not be the main component, but it still **offers a valuable directional estimate for the correction term**. The introduced variable $v$ represents how we exploit this valuable direction. In essence, **for $w > 1$, the second-order term provides valuable directional information.**
> > >
> > > To illustrate this, consider the Taylor expansion of $\cos x$ at $x = 0$: $\cos x = \sum_{0}^{\infty}\frac{(-1)^n}{(2n)!}x^{2n}$. **When $x = 0.2$**, $\cos x = 0.980067$. The real correction term is $\cos x - 1 = \sum_{1}^{\infty}\frac{(-1)^n}{(2n)!}x^{2n} = -0.0199334$, and the second-order term is $\frac{(-1)^1}{(2)!}x^{2} = -0.02$, where **the second-order term is the dominant component.**
> > >
> > > Similarly, **when $x = 1$,** $\cos x = 0.540302$, and the real correction term is $\cos x - 1 = \sum_{1}^{\infty}\frac{(-1)^n}{(2n)!}x^{2n} = -0.459698$, but **the second-order term is $\frac{(-1)^1}{(2)!}x^{2} = -0.5$, still making it the dominant part.**
> > >
> > > However, **when $x = 2$**, $\cos x = -0.416146$, and the real correction term is $\cos x - 1 = \sum_{1}^{\infty}\frac{(-1)^n}{(2n)!}x^{2n} = -1.416146$. Although **the second-order term is not accurate** in this case ($\frac{(-1)^1}{(2)!}x^{2} = -2$), **it consistently provides the same directional information, denoted by '-'.**
> > >
> > > **We acknowledge that there may be a more elegant explanation of our improvements, and we are committed to exploring and addressing these aspects in our future work.**
> > >
> > > ### Q2-2: sweeping for a range $v$.
> > >
> > > **We also conducted a sweeping analysis for variable $v$ over the range $0$ to $1$.** Specifically, when $v = 0$, it corresponds to the CFG scenario.  We set $w=3.0$.
> > >
> > > |  $v$  |  0| 0.25|  0.5      |   0.75 | 1.0| w=4.0,v=0 |
> > > |---- |:------:|:------:|:------:|:------:|:------:|:------:|
> > > |CLIP Score|26.12|26.37|26.54|26.64|**26.73**|26.34|
> > > |FID| 16.68 | **16.59** | 16.69 | 16.78 | 16.88| 18.18 |
> > >
> > > - **Consistent improvement** is observed in **CLIP Score**.
> > > - **FID exhibits improvement when $0 < v < 0.5$.**
> > > - Although a trade-off between CLIP Score and FID is present in the range $0.5 < v < 1$, **it is significantly more pronounced compared to CFG**. Specifically, CFG trades 1.5 FID loss for a 0.22 CLIP Score gain, while for $v = 1$, 2nd-order ICFG trades a 0.2 FID loss for a 0.61 CLIP Score gain.
> > >
> > > In summary, **the experiments demonstrate that the improvement of second-order ICFG compared to CFG is consistent and not sensitive to variations in the parameter $v$.**
> > > ### Q3: Experiments about the speedup.
> > >
> > > We acknowledge the existence of tuning for parameter $v$. However, it incurs a one-time cost that yields benefits for all subsequent samples.
> > >
> > > We conducted our experiments on an NVIDIA GeForce RTX 3090, using a batch size of 4. We performed 100 samplings to calculate the timings and utilized 10,000 images for the computation of FID and CLIP scores.
> > >
> > > |  Method  |   Time (seconds)| Extra Time Compared with CFG|  Estimation of U-Net Computation      |   FID |CLIP Score |
> > > |----------|:-------------:|:------:|:------:|:------:|:------:|
> > > | CFG |10.43 $\pm$ 0.23|**0%**| **100%** |    15.42 | 25.80 |
> > > | 2nd-order ICFG |15.01 $\pm$ 0.31|43.91%|  150% | **15.28** |**26.11** |
> > > | 0.2-0.8 2nd-order ICFG |13.17 $\pm$ 0.26| **26.27%**|**130%** |    **15.29** |**26.03** |
> > >
> > > * Due to the text encoder and VAE decoder, the real-time consumption of the 2nd-order ICFG is less than the estimated extra computation of the U-Net.
> > > * Through a preliminary selection of key timesteps (0.2-0.8) for applying the 2nd-order ICFG, **we achieve nearly full FID benefits and a 74% improvement in CLIP scores**, with a **reduced extra inference time of 26.27%**. We anticipate further enhancements in extra inference time by refining the selection of key timesteps.

---

> > > > ### Comment · Reviewer_LfUU · 2023-11-22
> > > >
> > > > Q2-1: I’m not sure how well the toy example illustrates the second order term importance. The second order term of course should provide more information when its estimation is accurate, but the authors also mentioned in the paper that there might be severe bias.
> > > >
> > > > Q2-2: That being said, I am convinced by the sweeping experiment results that the results are not sensitive to the choice of $v$. Nevertheless, it would be good to have some sort of heuristic to estimate a good choice of $v$.
> > > >
> > > > Q3: The quantitative result is reassuring.
> > > >
> > > > I have no further questions and would like to increase the score from 6 to 8.

---

> ### Author Response · Authors · 2023-11-23
>
> Dear reviewer LfUU,
>
> We express our heartfelt appreciation for your valuable review and advice. We firmly believe that with all your guidance, our manuscript has undergone significant improvement. We are committed to incorporating all discussions into the final version, and we are also considering the inclusion of some heuristics to estimate a suitable choice for $v$.
>
> Once again, we extend our sincere gratitude to reviewer LfUU.

---

### Official Review · Reviewer_cPdv · 2023-11-07

**Soundness:** 3 good
**Presentation:** 2 fair
**Contribution:** 2 fair
**Rating:** 5
**Confidence:** 3

**Summary:**

This work presents a new perspective on classifier-free guidance (CFG) for diffusion models imposing specific assumptions on the space of condition. Assuming that the condition space has a cone structure, the previous CFG can be seen as the first-order Taylor expansion of the proposed ICFG, and this work further presents a second-order ICFG that improves the Stable Diffusion model.

**Strengths:**

- The observation that there exists a mismatch between the transition distribution of the original forward process and the conditional forward process is new to the best of my knowledge.

- The idea that the mismatch is alleviated when assuming that the condition space has a cone structure is interesting.

**Weaknesses:**

- The contribution of the proposed ICFG is not clear: Although the authors state that the second-order ICFG introduces new valuable information, it is not clear which additional information it provides, and further the experimental results show a marginal improvement over previous CFG, e.g., FID improvement from 15.42 (CFG) to 15.22 (ICFG) and CLIP Score from 26.45 (CFG) to 26.86 (ICFG), on only single dataset (MS-COCO). Especially, ICFG does not seem to provide an improved balance between FID and CLIP Score. What is the main reason we should use ICFG instead of CFG?

- As the main motivation of this work is the mismatch between the transition kernels (in Theorem 3.1), this should be further analyzed, for example, how much difference in these kernels and how much it affects the generation quality. The proposed method should be evaluated after these validations.

- The assumptions (Assumptions 3.1 and 4.1) made to achieve the proposed method do not seem realistic and were not verified in the experiments.

**Questions:**

- What is the main reason we should use ICFG instead of CFG?

- How much does the enhanced transition kernel deviate from the original transition kernel (as in Theorem 3.1?) How much does the deviation affect the generation quality?

---

> ### Author Response · Authors · 2023-11-20
> **Rebuttal part I**
>
> We thank reviewer cPdv very much for recognizing the theoretical contributions and novelty of our idea. Your assistance in refining assumptions and conducting a deeper analysis of the gap between enhanced intermediate distributions has been particularly insightful.
>
> ### **W1-1: It is not clear which additional information the second-order term provides.**
>
> We apologize for any confusion in my previous wording. The second-order ICFG provides **more consistent enhanced intermediate distribution**s at various times, leading to improved overall performance. To elaborate further, the enhanced intermediate distributions at different time steps, denoted as $\bar{q}(\mathbf{x_{t_1} | c})$ and $\bar{q}(\mathbf{x_{t_2} | c})$ ($t_1 < t_2$), align more closely with the diffusion process ${\rm d}\mathbf{x} = \mathbf{f}(\mathbf{x},t) {\rm d} t + g(t) {\rm d} \mathbf{w}$.  In this way, the original diffusion reverse process can be more easily accommodated for the conditional reverse diffusion process due to the reduced mismatch of the enhanced intermediate distributions.
>
>
> ### **W1-2: The improvement of second-order ICFG is not clear.**
>
> We apologize for any confusion in our experiment descriptions. While maintaining **the same training policy as CFG**, the continuous feature of the condition space enables second-order ICFG to **consistently outperform** CFG in both FID and CLIP scores across various CFG strengths ($w$).
>
> For a clearer understanding of our advancements in generation quality, consider the comparison between second-order ICFG with $w = 2.0, v = 0.25, C=C_{all}$ and CFG with $w = 3.0$ (depicted as the second orange spot and the third blue spot in Figure 2). **Despite nearly identical CLIP scores (26.12 and 26.11), we observed a notable improvement of $(16.68 - 15.28)/16.68 =$ 8.3% in FID**. This implies that with increased control strength, the generated quality is significantly enhanced.
>
> It is noteworthy that the **CLIP Score improvement achieved by our second-order ICFG is nearly half of the overall improvements observed from Stable Diffusion v1.1 to v1.5** ([comparison chart](https://huggingface.co/CompVis/stable-diffusion/resolve/main/v1-1-to-v1-5.png)). This indicates a substantial improvement in performance.
>
> Due to GPU limitations, we were unable to train SD from scratch. However, we are currently training **another framework, U-ViT**, with a resolution of 256x256 on the COCO dataset from scratch to fully explore the capabilities of our ICFG. The preliminary results are listed below, and we plan to update them upon completion of the training.
>
> | Training Policy \FID \ Training Steps| 5w | 10w | 15w | 20w | 25w | 30w | 35w | 40w | 45w | 50w| 55w | 60w | 65w | 70w | 75w | 80w |
> |----------|:-------------:|------:|--:|--:|--:|--:|--:|--:|--:|--:|--:|--:|--:|--:|--:|--:|
> | Original | 34.23 | 13.64 | 11.26 | 10.38 | 9.78 | 8.98 | 8.98 | 8.76 | 8.58 | 8.52 | 8.37 | 8.37 | 8.27 | 8.32 | 8.39 | 8.10 |
> | ICFG | 24.69 | 13.51 | 11.00 | 10.13 | 9.69 | 9.09 | 8.82 | 8.68 | 8.54 | 8.41 | 8.35 | 8.21 | 8.29 | 8.15 | 8.10 | **7.92** |
>
> We plan to report the final FID and CLIP scores after completing the entire training process. As of now, we have observed a notable **3% improvement** in FID with  **ICFG training algorithm and CFG sampling algorithm** compared with the original training policy.
>
> ### **W1-3: The main reason we should use ICFG instead of CFG.**
>
> We apologize for any confusion, and we want to clarify that we have made significant updates to our manuscript to enhance clarity regarding the assumptions and calculation of the mismatch. Following a thorough rebuttal, we will incorporate all additional experiments into the manuscript.
>
> While CFG has achieved notable success in various impactful projects, our focus is on elucidating **potential theoretical inconsistencies**. This inconsistency serves as the motivation behind our ICFG, and we present both theoretical and experimental analyses of the problem.
>
> Theoretically, we provide a detailed analysis and present policies for addressing the issue on both the training and sample sides. Our proposed ICFG in the condition space effectively mitigates the distribution mismatch problem.
>
> Experimentally, for integration into existing Stable Diffusion, we introduce a practical approach, the second-order ICFG. This modification requires only a minor adjustment involving the addition of a second term, without the need for additional training. Notably, this approach consistently improves both FID and CLIP scores. **In scenarios with high CLIP scores, we observe an FID improvement of approximately 8%**. Additionally, we have undertaken efforts to optimize the speed of our approach, aiming to make it more widely applicable and accepted within the community.

---

> ### Author Response · Authors · 2023-11-20
> **Rebuttal part II**
>
> ### **W2-1: The differences in the transition kernels.**
>
> The computation of transition kernels may pose challenges, but calculating the two distinct enhanced intermediate distributions resulting from differences in transition kernels is more straightforward.
>
> We provide a theoretical analysis of the ratio between these two types of enhanced intermediate distributions. One approach involves obtaining the conditional $x_0$ and then disturbing it, while the other involves acquiring the disturbed $x_t$ and subsequently applying the condition.
>
> In common scenarios, where we have $N$ paired $(\mathbf{x}_0^i, \mathbf{c}^i), i = 1, ..., N$ training data, given $\mathbf{c}=\mathbf{c}^1$, **the ratio of the probabilities of the two enhanced intermediate distributions at $\mathbf{x}_t$ is expressed as:**
>
> $$
> \frac{Z_t}{Z_0}\left[\frac{\sum_{i = 1}^{N}e^{\|-\mathbf{x}_t - \alpha_t\mathbf{x}_0^i\|^2/(\beta_t^2)} }{e^{\|-\mathbf{x}_t - \alpha_t\mathbf{x}_0^1\|^2/(\beta_t^2)}}\right]^w,
> $$
>
> where $Z_t$ and $Z_0$ are constants. Additional calculation details can be found in Appendix A.1.
>
> In a common scenario, when $\mathbf{x_t}$ is approximately equidistant between $\alpha_t \mathbf{x}_0^1$ and $\alpha_t \mathbf{x}_0^2$ with $w\ne 0$, the ratio is evidently not 1 and becomes highly sensitive to changes in $\mathbf{x_t}$.
>
> ### **W2-2: How much difference how it affect the generation quality?**
>
> To ensure a fair evaluation of generation quality, we compare FIDs at the same CLIP Score. **Our training-free second-order ICFG demonstrates a maximum improvement of 8.3% in FID**. Further details can be found in the response to W1-2.
>
> ### **W3: The assumptions (Assumptions 3.1 and 4.1) made to achieve the proposed method do not seem realistic and were not verified in the experiments.**
>
> We appreciate your understanding, and we would like to clarify Assumptions 3.1 and 4.1 for better comprehension.
>
> In practice, we utilize the CLIP embedding space as the condition space, which is 77x768-dimensional. To construct a cone structure, we use a finite number of outputs from CLIP as the basis, extending it by multiplying it with any positive number.
>
> **The true assumption, in essence, is that there are no identical directions among the outputs of CLIP for different texts,  which is more realistic**. Importantly, Assumption 3.1 aligns with CFG. While CFG operates in the score domain (output of U-Net), ICFG applies the cone assumption within the condition space.
>
> The challenge posed by Assumption 3.1 lies in adapting the score predictor (U-Net for Stable Diffusion) because, during training, the U-Net has never encountered the extended cone structure. However, owing to the continuity of the extended condition space, the unadapted U-Net performs well within a small range of the cone structure. Consequently, the approximated second-order ICFG with the unadapted U-Net still demonstrates a notable improvement compared to CFG.
>
> While Assumption 4.1 may not always hold for uniformly bounded $M_n$, it serves as a conceptual aid. **In reality, the bounds need not be uniform and can be extended, for instance, to $M_{n+1} \sim o\left(\sqrt{n+1}\left[\frac{n+1}{eB}\right]^{n+1}\right)$, where values like $M_{n+1}= 10^n$ are acceptable.** This extension provides a more practical perspective. Additional details can be found in Appendix B.
>
> ### **Q1: What is the main reason we should use ICFG instead of CFG?**
> Referring to W1-3.
>
> ### **Q2: How much does the enhanced transition kernel deviate from the original transition kernel (as in Theorem 3.1?) How much does the deviation affect the generation quality?**
> Referring to W2.

---

> ### Author Response · Authors · 2023-11-23
>
> Dear reviewer cPdv,
>
> Thank you for your insightful review. We have thoroughly considered your concerns, particularly those related to additional experiments, the advantages of ICFG, refining assumptions, and conducting a more in-depth analysis of the gap between enhanced intermediate distributions. In our response, we have endeavored to address these issues to the best of our ability.
>
> As the discussion stage nears its conclusion, we welcome any further questions or suggestions you may have. We are eager to engage in further discussion and address any remaining points of consideration.

---

> > ### Comment · Reviewer_cPdv · 2023-11-23
> > **Thank you for the response**
> >
> > Thank you for the detailed response.
> >
> > - I do not understand the performance comparison made in the response of *W1-2*: Why is the performance improvement compared with different $w$? When comparing CFG and ICFG with $w=3.0$, the CLIP scores are almost the same but the FID of ICFG seems to be worse than CFG. Further when comparing with $w=2.0$, the FID is almost the same while the CLIP score of ICFG has a marginal improvement over CFG.
> >
> > - Furthermore, when comparing the best performance of CFG and ICFG, ICFG does not provide clear improvement over CFG, achieving almost the same FID and CLIP score.
> >
> > - Although the motivation of "elucidating potential theoretical inconsistencies" is interesting, the analysis or the experimental results for this is not sufficient. The results in Table 1 do not show clear improvements over CFG and further the improvement claimed by the authors (8%) is not made in a fair setting. There should be more experimental results on other datasets to strengthen the author's claim.
> >
> > - The response of *W3* does not justify applying the cone assumption for realistic setting. I recommend the authors from avoiding overclaim such as ICFG demonstrates "remarkable improvement compared to CFG" which is not validated through experiments.

---

> > > ### Author Response · Authors · 2023-11-23
> > >
> > > Dear Reviewer cPdv,
> > >
> > > We acknowledge and understand your concerns, and we have made significant efforts to address them. Our solutions include **explaining why comparing CFG and ICFG with the same CLIP Score** ( The **CLIP Score serves as an effective metric** for evaluating text-image alignment, essentially **capturing the real control strength achieved by the algorithm**), **incorporating standard deviation to showcase the stability of ICFG's advantages over CFG**, and **presenting visual results on other models and datasets**. Furthermore, we provide an explanation for the application of the cone assumption within a limited range and the use of Taylor expansion to extend its applicability. We trust that our detailed explanation can help alleviate your concerns.
> > >
> > > As the discussion stage nears its conclusion, we welcome any further questions or suggestions you may have.

---

> > > > ### Comment · Reviewer_cPdv · 2023-11-23
> > > >
> > > > Thank you for the detailed response.
> > > >
> > > > - As the authors acknowledge, comparing the results with the same $w$, the improvement is marginal and the best results achieved by ICFG are almost the same as CFG.
> > > >
> > > > - Now I understand why the authors were comparing with the same CLIP score, but to validate the author's claim, further quantitative results on other datasets should be provided. I appreciate the visualization in Figure 5 of Appendix F.
> > > >
> > > > - I do not understand the statement "if the space can be extended to form a cone". How can we determine if it can be extended for the experiments? Should it be checked by analyzing the results applying ICFG?
> > > >
> > > > As some of my concerns are addressed, I raise my score from 3 to 5, but I lean towards rejection.

---

> ### Author Response · Authors · 2023-11-23
>
> Thank you sincerely for your valuable feedback. We also acknowledge and apologize for any confusion resulting from Table 1. We hope the following explanation will be helpful.
> ### Q1: About Table 1 Experiments.
>
> For a clearer understanding of our improvements, we summarize key points here, and trust that the subsequent explanation will provide further clarity.
>
> * When $v=0$, it indicates CFG, and when $v = 0.25$, it indicates ICFG settings.
> * For **FID, lower values indicate better performance**.
> * For **CLIP Score, larger values indicate better performance**.
> * We conducted 5 times of sampling to ensure stable results for our ICFG.
>
> |  Method  |      w=2.0,v=0;    |  w=2.0, v=0.25; | w=3.0, v = 0;| w = 3.0, v=0.25 |
> |----------|:-------------:|:------:|:------:|:------:|
> | FID |  15.42 |    $15.25\pm0.03$ | 16.68 |$16.58\pm0.04$ |
> | CLIP Score |  25.80 |    $26.13\pm0.02$ | 26.12 |$26.36\pm0.02$|
>
> ### Q1-1: Compare with same $w$.
> In comparing the first two columns, with same $w=2$, FID of ICFG (15.25) is better than CFG(15.42); CLIP Score of ICFG(26.13) is better than CFG(25.80).
>
> Similarly, in comparing the last two columns, with same $w=3$, FID of ICFG (16.58) is better than CFG(16.68); CLIP Score of ICFG(26.36) is better than CFG(26.12).
>
> In summary of the Table 1 experiments, for all $w=1,2,3,4,5$, the FID and CLIP Score of ICFG consistently outperform those of CFG.
>
> While we acknowledge that the improvement may appear marginal when comparing the same $w$, we attribute this to enhancements in both text-image alignment (CLIP Score) and generated quality (FID).
>
> To provide a more comprehensive comparison, we introduce the following analysis. **The motivation behind CFG and ICFG is to achieve enhanced control strength, as evaluated by the CLIP Score**, which measures the alignment between the control condition text and the generated image. Our aim is to assess generation quality under the constraint of the same text-image alignment. Therefore, **we conduct a comparison with the same CLIP Score to fairly evaluate the improvement of generation quality itself.**
>
> In summary, when referring to **the same $w$,** we are essentially comparing **the desired control strength in the input**. However, the evaluation of **whether the control strength is sufficient being determined by the CLIP Score**. This is the reason why we conduct the following experiments.
> ### Q1-2: Compare with same CLIP Score.
>
> Comparing the middle two columns, with almost the same CLIP Score (26.11), the decrease in our FID is $(16.68 - 15.28)/16.68 = 8.3$%, which is substantial. We apologize for any subjective expressions and leave it to the reader to assess the improvement.
> ### Q1-3: More experiments.
>
> We conduct our experiments on another fine-tuned model, anything-V4 ([https://huggingface.co/xyn-ai/anything-v4.0](https://huggingface.co/xyn-ai/anything-v4.0)). It is evident from the images that second-order ICFG outperforms CFG, producing images with better-rendered hands and closer alignment to the provided prompts, and the results are visualized in Figure 5 of Appendix F.
> ### Q2: The response of _W3_ does not justify applying the cone assumption for realistic setting.
>
> Our previous response emphasizes that the cone is typically constructed by extending the original conditional space. The crucial factor is whether the score predictor can adapt to it. If the space is continuous, it usually works within a small range. **From our perspective, if the space can be extended to form a cone, the ICFG should be adaptable within a limited range.**
>
> **To assess the U-Net's ability to capture the cone structure, we employ two complementary approaches.** Firstly, we directly apply the guidance strength on the condition space to observe the generated results, as demonstrated in Appendix E. Secondly, we calculate the FID and CLIP Score of the samples as metrics for evaluation.

---

> ### Author Response · Authors · 2023-11-23
>
> We thank you very much for the invaluable reviews and suggestions.
>
> * The improvement is stable.
>
> * We will include more validations on the comparison with same CLIP Score.
>
> * We can check them by FIDs of same CLIP Score.

---

### Author Response · Authors · 2023-11-20
**Rebuttal to all reviewers**

We express our sincere gratitude to all the reviewers for their invaluable contributions.

We thank reviewer cPdv very much for recognizing the theoretical contributions and novelty of our idea. Their assistance in refining assumptions and conducting a deeper analysis of the gap between enhanced intermediate distributions has been particularly insightful.

We extend our appreciation to Reviewer LfUU for acknowledging both our theoretical analysis and empirical results. Their guidance in accelerating our method and perfecting the non-strict 2nd order term reasons has been tremendously helpful.

We are deeply thankful for Reviewer jYKn's recognition of our novel formulation and well-studied hyper-parameters. We sincerely value their guidance in recommending the inclusion of additional empirical results, particularly in the form of visual comparisons.

Our gratitude extends to Reviewer UdAr for acknowledging the novelty in our approach to combining guidance strength and condition. Their assistance in correcting definitions, perfecting proofs, and the formulation of the theorem, along with their guidance in including additional empirical results and speeding up our method, has been invaluable.

Thank you all very much again for your thoughtful and constructive feedback.

---

### Meta-Review · Area_Chair_2eC8 · 2023-12-11

**Metareview:**

This paper proposes a generalization of the classifier-free guidance technique for diffusion models, that imposes a structure on the continuous condition space, dubbed as inner classifier-free guidance (ICFG). The authors further proposes a second-order Taylor expansion of ICFG, and show that the original CFG can be regarded as a first-order IFCG. The experimental validation of the proposed ICFG with the StableDiffusion model shows that the it enhances the fidelity-diversity tradeoff.

The paper received mostly positive reviews except for one reviewer who gave it the rating of borderline reject. The reviewers considered the motivation coming from the observation of the mismatch between the original and the conditional forward process as new, and the proposed method of imposing structural assumptions into the condition space as also novel and interesting. They also found the paper well-written.

However, there were some concerns on the weak performance of the proposed ICFG, which has marginal performance gain over the conventional CFG, and even that performance gain is achievable with very careful hyperparameter tuning. The reviewers believe that the practical impact of the proposed method could be low, considering that the proposed ICFG results in increased sampling time. While the authors addressed away some of the concerns such as the sensitivity to the hyperparameters in the rebuttal, most of the reviewers considered the paper as borderline due to the weak performance gain.

Yet, as mentioned in the reviews, even with limited practical impact, this paper may provide new insights to the researchers working on conditional diffusion models, which could further promote research in this direction. Thus, I believe that the paper is worth sharing to a broad audience at ICLR 2024.

**Justification For Why Not Higher Score:**

The performance gain obtainable by the proposed ICFG method is weak compared to the conventional CFG, which may result in very little practical impact considering the increased sampling time.

**Justification For Why Not Lower Score:**

The paper provides new insights on the mismatch between the unconditional and conditional forward processes, as well as a novel structured constraint into the conditional space.

---

### Decision · Program_Chairs · 2024-01-16

Accept (poster)